# From Trainable Negative Depth to Edge Heterophily in Graphs

**Yuchen Yan[1], Yuzhong Chen[2], Huiyuan Chen[2], Minghua Xu[2],**
**Mahashweta Das[2], Hao Yang[2], Hanghang Tong[1]**
[1]University of Illinois at Urbana Champaign, IL, USA
[2]Visa Research, CA, USA
{yucheny5, htong}@illinois.edu
{yuzchen, hchen, mixu, mahdas, haoyang}@visa.com

## Abstract

Finding the proper depth $d$ of a graph convolutional network (GCN) that provides strong representation ability has drawn significant attention, yet nonetheless largely remains an open problem for the graph learning community. Although noteworthy progress has been made, the depth or the number of layers of a corresponding GCN is realized by a series of graph convolution operations, which naturally makes $d$ a positive integer ($d \in \mathbb{N}+$). An interesting question is whether breaking the constraint of $\mathbb{N}+$ by making $d$ a real number ($d \in \mathbb{R}$) can bring new insights into graph learning mechanisms. In this work, by redefining GCN's depth $d$ as a trainable parameter continuously adjustable within $(-\infty, +\infty)$, we open a new door of controlling its signal processing capability to model graph homophily/heterophily (nodes with similar/dissimilar labels/attributes tend to be inter-connected). A simple and powerful GCN model TEDGCN, is proposed to retain the simplicity of GCN and meanwhile automatically search for the optimal $d$ without the prior knowledge regarding whether the input graph is homophilic or heterophilic. Negative-valued $d$ intrinsically enables high-pass frequency filtering functionality via augmented topology for graph heterophily. Extensive experiments demonstrate the superiority of TEDGCN on node classification tasks for a variety of homophilic and heterophilic graphs.

## 1 Introduction

Graph convolutional network (GCN) [28, 50, 19, 47] has exhibited superb power in a myriad of graph learning tasks, such as knowledge graph reasoning [56, 53, 54, 52, 36], network alignment [63, 64, 68, 74, 20, 35, 67], multi-network mining [65, 73, 42, 13, 27, 17], networked time series imputation [51, 25, 55, 16], node recommendation [26, 2, 3], and many more. Since the representation capability of GCN is largely determined by its depth, i.e., the number of graph convolution layers, tremendous research efforts have been made on finding the optimal depth that strengthens the model's ability for downstream tasks. Upon increasing the depth, the over-smoothing issue arises: a GCN's performance is deteriorated if its depth exceeds a certain threshold [28]. It is found in [32] that a graph convolution operation is a special form of Laplacian smoothing [48]. Thus, the similarity between the graph node embeddings grows with the depth so that these embeddings eventually become indistinguishable. Various techniques have been developed to alleviate this issue, e.g., applying pairwise normalization can make distant nodes dissimilar [69], and dropping sampled edges during training slows down the growth of embedding smoothness as the depth increases [43].

Other than the over-smoothing issue due to a large GCN depth, another fundamental phenomenon widely existing in real-world graphs is homophily and heterophily. In a homophilic graph, nodes with

37th Conference on Neural Information Processing Systems (NeurIPS 2023).

similar labels or attributes tend to be inter-connected, while in a heterophilic graph, connected nodes usually have distinct labels or dissimilar attributes. Most graph neural networks (GNNs) are developed based on the homophilic assumption [66], while models able to perform well on heterophilic graphs often need special treatment and complex designs [4, 76, 62]. Despite the achievements made by these methodologies, little correlation has been found between the adopted GNN model's depth and its capability of characterizing graph heterophily.

For almost any GNN model, the depth needs to be manually set as a hyper-parameter before training, and finding the proper depth usually requires a considerable amount of trials or good prior knowledge of the graph dataset. Since the depth represents the number of graph convolution operations and naturally takes only positive integer values, little attention has been paid to the question of whether a non-integer depth is realizable, and if yes, whether it is practically meaningful, and what kind of unique advantages it can bring to current graph learning mechanisms.

This work revisits the GCN depth from spectral and spatial perspectives and explains the intrinsic connections between the following key ingredients in graph learning: (i) the depth of a GCN, (ii) the spectrum of the graph signal, and (iii) the homophily/heterophily of the underlying graph. Firstly, through eigen-decomposition of the symmetrically normalized graph Laplacian, we present the connection between graph homophily/heterophily and the eigenvector frequencies. Secondly, by introducing the concept of eigengraph, we show the graph topology is equivalent to a weighted linear combination of eigengraphs, and the weights determine the GCN's capability of capturing homophilic/heterophilic graph signals. Thirdly, we reveal that the eigengraph weights can be controlled by GCN's depth, so that an automatically tunable depth parameter is needed to adjust the eigengraph weights into the designated distribution in match of the underlying graph homophily/heterophily.

To realize the adaptive GCN depth, we extend its definition from a positive integer to an arbitrary real number with theoretical feasibility guarantees from functional calculus [45]. With a trainable depth parameter, we propose a simple and powerful model, Trainable Depth-GCN (TEDGCN), with two variants. Extensive experiments demonstrate TEDGCN's ability of automatically searching for the optimal depth, and it is found that negative-valued depth plays the key role in handling heterophilic graphs. Systematical investigation on the optimal depth is conducted via various real-world and synthetic datasets. It in turn inspires the development of a novel graph augmentation methodology. With clear geometric interpretability, the augmented graph structure possesses supreme advantages over the raw input topology, especially for graphs with heterophily. The main contributions of this paper are summarized as follows:

- **Analysis.** The intrinsic relationship between the negative GCN depth and graph heterophily is discovered; In-depth geometric and spectral explanations are presented.

- **Problem.** A novel problem of automatic GCN depth tuning for graph homophily/heterophily detection is formulated. To our best knowledge, this work presents the first trial to make GCN's depth negative and trainable by redefining it on the real number domain.

- **Model.** A simple and powerful model TEDGCN with two variants (TEDGCN-S and TEDGCN-D) is proposed. A novel graph augmentation method is discussed.

- **Experiments.** Our model achieves superior performance on semi-supervised node classification tasks on 11 graph datasets.

## 2 Preliminaries

**Notations.** We utilize bold uppercase letters for matrices (e.g., $\mathbf{A}$), bold lowercase letters for column vectors (e.g., $\mathbf{u}$) and lowercase letters for scalars (e.g., $\alpha$). We use the superscript $\top$ for the transpose of matrices and vectors (e.g., $\mathbf{A}^\top$ and $\mathbf{u}^\top$). An attributed undirected graph $\mathcal{G} = \{\mathbf{A}, \mathbf{X}\}$ contains an adjacency matrix $\mathbf{A} \in \mathbb{R}^{n \times n}$ and an attribute matrix $\mathbf{X} \in \mathbb{R}^{n \times q}$ with the number of nodes $n$ and the dimension of node attributes $q$. $\mathbf{D}$ denotes the diagonal degree matrix of $\mathbf{A}$. The adjacency matrix with self-loops is given by $\tilde{\mathbf{A}} = \mathbf{A} + \mathbf{I}$ ($\mathbf{I}$ is the identity matrix), and all variables derived from $\tilde{\mathbf{A}}$ are decorated with symbol $\tilde{\phantom{x}}$, e.g., $\tilde{\mathbf{D}}$ represents the diagonal degree matrix of $\tilde{\mathbf{A}}$. $\mathbf{M}^d$ stands for the $d$-th power of matrix $\mathbf{M}$, while the parameter and node embedding matrices in the $d$-th layer of a GCN are denoted by $\mathbf{W}^{(d)}$ and $\mathbf{H}^{(d)}$.

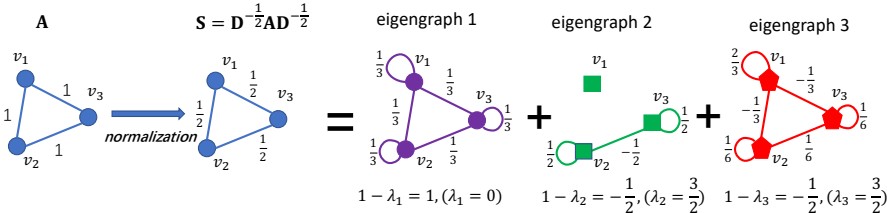

Figure 1: Decompose the symmetrically normalized adjacency matrix into three eigengraphs.

**Graph convolutional network (GCN) and simplified graph convolutional network (SGC).** The layer-wise message-passing and aggregation of GCN [28] is given by

$$\mathbf{H}^{(d+1)} = \sigma(\tilde{\mathbf{D}}^{-\frac{1}{2}}\tilde{\mathbf{A}}\tilde{\mathbf{D}}^{-\frac{1}{2}}\mathbf{H}^{(d)}\mathbf{W}^{(d)}), \tag{1}$$

where $\mathbf{H}^{(d)}/\mathbf{H}^{(d+1)}$ stands for the embedding matrix ($\mathbf{H}^{(0)} = \mathbf{X}$) in the $d$-th/$(d+1)$-th layer; $\mathbf{W}^{(d)}$ is the trainable parameter matrix; and $\sigma(\cdot)$ is the non-linear activation function. With $\sigma(\cdot)$ removed in each layer, SGC [59] is obtained as below:

$$\mathbf{H}^{(d)} = \tilde{\mathbf{S}}^d\mathbf{X}\mathbf{W}, \tag{2}$$

where $\tilde{\mathbf{S}} = \tilde{\mathbf{D}}^{-\frac{1}{2}}\tilde{\mathbf{A}}\tilde{\mathbf{D}}^{-\frac{1}{2}}$, and the parameter of each layer $\mathbf{W}^{(i)}$ are compressed into one trainable $\mathbf{W} = \prod_{i=0}^{d-1} \mathbf{W}^{(i)}$.

**Graph Laplacian and spectrum.** In graph theory, graph Laplacian $\mathbf{L} = \mathbf{D} - \mathbf{A}$ and its symmetrically normalized correspondence $\mathbf{L}_{sym} = \mathbf{I} - \mathbf{D}^{-\frac{1}{2}}\mathbf{A}\mathbf{D}^{-\frac{1}{2}}$ possess critical properties of the underlying graph $\mathcal{G}$. $\mathbf{L}_{sym}$ has eigenvalues $[\lambda_1, \lambda_2, \ldots, \lambda_n]$, where $\lambda_i \in [0, 2), \forall i \in \{1, 2, \ldots, n\}$ [11].[1] Here, they are put in the ascending order: $0 = \lambda_1 \le \lambda_2 \le \cdots \le \lambda_n < 2$. It can be eigen-decomposed as: $\mathbf{L}_{sym} = \mathbf{U}\mathbf{\Lambda}\mathbf{U}^\top$, where $\mathbf{U} = [\mathbf{u}_1, \mathbf{u}_2, \ldots, \mathbf{u}_n]$ is the eigenvector matrix ($\mathbf{u}_i \perp \mathbf{u}_j, \forall i \ne j$), and $\mathbf{\Lambda}$ is the diagonal eigenvalue matrix. For each eigenvector $\mathbf{u}_i$, we have $\mathbf{u}_i\mathbf{u}_i^\top \in \mathbb{R}^{n \times n}$. As we will show in Section 3, this $n \times n$ matrix can be viewed as the weighted adjacency matrix of a graph with possible negative edges, which we name as the $i$-th eigengraph of $\mathcal{G}$. Accordingly, $\mathbf{L}_{sym}$ can be written as the linear combination of all eigengraphs weighted by the corresponding eigenvalues [11]:

$$\mathbf{L}_{sym} = \lambda_1\mathbf{u}_1\mathbf{u}_1^\top + \ldots + \lambda_i\mathbf{u}_i\mathbf{u}_i^\top + \ldots + \lambda_n\mathbf{u}_n\mathbf{u}_n^\top, \tag{3}$$

where the first eigenvalue $\lambda_1 = 0$ [46]. Thus, for SGC, we have

$$\tilde{\mathbf{S}} = \mathbf{I} - \tilde{\mathbf{L}}_{sym} = \tilde{\mathbf{U}}(\mathbf{I} - \tilde{\mathbf{\Lambda}})\tilde{\mathbf{U}}^\top = \sum_{i=0}^{n}(1 - \tilde{\lambda}_i)\tilde{\mathbf{u}}_i\tilde{\mathbf{u}}_i^\top. \tag{4}$$

A SGC with $d$ layers requires $d$ consecutive graph convolution operations, which involves the multiplication of $\tilde{\mathbf{S}}$ by $d$ times. Due to the orthogonality of $\tilde{\mathbf{U}}$, namely, $\tilde{\mathbf{U}}^\top\tilde{\mathbf{U}} = \mathbf{I}$, we obtain

$$\tilde{\mathbf{S}}^d = \tilde{\mathbf{U}}(\mathbf{I} - \tilde{\mathbf{\Lambda}})\tilde{\mathbf{U}}^\top\tilde{\mathbf{U}}(\mathbf{I} - \tilde{\mathbf{\Lambda}})\tilde{\mathbf{U}}^\top \ldots \tilde{\mathbf{U}}(\mathbf{I} - \tilde{\mathbf{\Lambda}})\tilde{\mathbf{U}}^\top = \tilde{\mathbf{U}}(\mathbf{I} - \tilde{\mathbf{\Lambda}})^d\tilde{\mathbf{U}}^\top = \sum_{i=1}^{n}(1 - \tilde{\lambda}_i)^d\tilde{\mathbf{u}}_i\tilde{\mathbf{u}}_i^\top, \tag{5}$$

where $1 - \tilde{\lambda}_i \in (-1, 1]$, and the depth $d$ of SGC serves as the power of $\tilde{\mathbf{S}}$'s eigenvalues. $\tilde{\mathbf{S}}^d$ can be viewed as the sum of eigengraphs $\tilde{\mathbf{u}}_i\tilde{\mathbf{u}}_i^\top$ weighted by coefficients $(1 - \tilde{\lambda}_i)^d$.

**Graph homophily and heterophily.** Various homophily/heterophily measures are developed to capture how edges tend to link nodes with the same labels and similar attributes, such as edge homophily[1, 76, 75, 37], node homophily[41], and class homophily[34]. Without the loss of generality, we focus on one mostly used homophily metric, the edge homophily: $h(\mathcal{G}) = \frac{\sum_{i,j,\mathbf{A}[i,j]=1}\langle\mathbf{y}[i]=\mathbf{y}[j]\rangle}{\sum_{i,j}\mathbf{A}[i,j]} \in [0, 1]$, where $\langle x \rangle = 1$ if $x$ is true and 0 otherwise. A graph is more homophilic for $h(\mathcal{G})$ closer to 1 or more heterophilic for $h(\mathcal{G})$ closer to 0.

---

[1]This work focuses on connected graph without bipartite components (i.e., a connected component which is a bipartite graph).

## 3 Model

**Overview.** Firstly, we establish the intrinsic relationship between graph spectrum and graph heterophily. Secondly, we show how the positive/negative depth $d$ of a GCN affects the eigengraph weights which in turn impacts the algorithm's capability to capture homophilic/heterophilic graph signals. Thirdly, with the help of functional calculus [45], we present the theoretical feasibility of extending the domain of $d$ from $\mathbb{N}+$ to $\mathbb{R}$. Finally, by making $d$ a trainable parameter, we present our model TEDGCN and its variants, which are capable of automatically detecting the homophily/heterophily of the input graph and finding the optimal depth.

**Eigenvector frequency and entry deviation.** The frequency of graph Laplacian eigenvector $\mathbf{u}_i$ reflects how much the $j$-th entry $\mathbf{u}_i[j]$ deviates from the $k$-th entry $\mathbf{u}_i[k]$ for each connected node pair $v_j$ and $v_k$ in $\mathcal{G}$ [46]. This deviation is measured by the set of zero crossings of $\mathbf{u}_i$: $\mathcal{Z}(\mathbf{u}_i) := \{e = (v_j, v_k) \in \mathcal{E} : \mathbf{u}_i[j]\mathbf{u}_i[k] < 0, j \neq k\}$, where $\mathcal{E}$ is the set of edges in graph $\mathcal{G}$. Larger/smaller $|\mathcal{Z}(\mathbf{u}_i)|$ indicates higher/lower eigenvector frequency. A zero-crossing also corresponds a negative weighted edge in an eigengraph. Due to the widely existing positive correlation between $\lambda_i$ and $|\mathcal{Z}(\mathbf{u}_i)|$ [46], large/small eigenvalues mostly correspond to the high/low frequencies of the related eigenvectors. As illustrated by the toy example of $n = 3$ in Figure 1, for $\lambda_1 = 0$, we have $|\mathcal{Z}(\mathbf{u}_1)| = 0$, and eigengraph $\mathbf{u}_1\mathbf{u}_1^\top$ is fully-connected; negative edge weights exist in the 2nd and 3rd eigengraphs, indicating more zero crossings ($|\mathcal{Z}(\mathbf{u}_2)| = 1$ and $|\mathcal{Z}(\mathbf{u}_3)| = 2$) and higher eigenvector frequencies.

**Entry deviation and graph heterophily.** Reflected by entry deviations, eigenvector frequency and zero-crossings determine the number of negatively weighted edges ($\mathbf{u}_i[j]\mathbf{u}_i[k] < 0$, $j \neq k$) in a corresponding eigengraph $\mathbf{u}_i\mathbf{u}_i^\top$. For a GCN/SGC, conducting message-passing along negatively weighted edges makes the embeddings of the connected node pairs $(v_j, v_k)$ dissimilar. Naturally, higher embedding dissimilarity is more likely to render different label predictions between the node pair, and this generates heterophily. Thus, eigenvectors with higher frequencies and accordingly higher entry deviations contribute more to heterophilic graph signals, and vice versa.

**Graph heterophily and GCN's depth.** Based on above analyses, high frequency eigenvectors and their corresponding eigengraphs have advantage on capturing graph heterophily. A graph neural network should be able to give high frequency eigengraphs larger weights when modeling heterophilic graphs, whereas low frequency ones should carry larger weights when dealing with homophilic graphs. The weights of eigengraphs are controlled by GCN/SGC's depth $d$ as follows. For a SGC of depth $d$, the weight of the $i$-th eigengraph is $(1 - \tilde{\lambda}_i)^d$, and changing the depth $d$ of SGC adjusts the weights of different eigengraphs. Therefore, depth $d$ controls the model's capability of effectively filtering low/high-frequency signals for graph homophily/heterophily.

**Redefined GCN's depth.** A question is naturally raised: instead of manually setting the depth $d$, can $d$ be built into the model as a trainable parameter? If yes, a proper set of the eigengraph weights matching the graph homophily/heterophily can be automatically reached by finding the optimal $d$ in an end-to-end fashion during training.

Differentiable variables need continuity, which requires the extension of depth $d$ from the discrete positive integer domain ($\mathbb{N}+$) to the continuous real number domain $\mathbb{R}$. According to functional calculus [45], applying an arbitrary matrix function $f$ on a graph Laplacian $\mathbf{L}_{sym}$ is equivalent to applying the same function only on the eigenvalue matrix $\mathbf{\Lambda}$:

$$f(\mathbf{L}_{sym}) = \mathbf{U}f(\mathbf{\Lambda})\mathbf{U}^\top = \mathbf{U} \begin{bmatrix} f(\lambda_1) & \cdots & 0 \\ \vdots & \ddots & \vdots \\ 0 & \cdots & f(\lambda_n) \end{bmatrix} \mathbf{U}^\top, \tag{6}$$

which also applies to $\tilde{\mathbf{L}}_{sym}$ and $\tilde{\mathbf{S}}$. Armed with this, we seek to realize an *arbitrary* depth SGC via a power function as $f(\tilde{\mathbf{S}}) = \tilde{\mathbf{S}}^d = \tilde{\mathbf{U}}(\mathbf{I} - \tilde{\mathbf{\Lambda}})^d\tilde{\mathbf{U}}^\top = \sum_{i=1}^n (1 - \tilde{\lambda}_i)^d\tilde{\mathbf{u}}_i\tilde{\mathbf{u}}_i^\top (d \in \mathbb{R})$. However, since $\tilde{\lambda}_i \in [0, 2)$, we have $(1 - \tilde{\lambda}_i) \leq 0$ when $1 \leq \tilde{\lambda}_i < 2$, and for $(1 - \tilde{\lambda}_i)$ taking zero or negative values, $(1 - \tilde{\lambda}_i)^d$ is not well-defined or involving complex-number-based calculations for a real-valued $d$ (e.g.,$(-0.5)^{\frac{3}{8}}$) [45]. Moreover, even for integer-valued $d$s under which $(1 - \tilde{\lambda}_i)^d$ is easy to compute, the behavior of $(1 - \tilde{\lambda}_i)^d$ is complicated versus $\tilde{\lambda}_i$ and diverges when $\tilde{\lambda}_i = 1$ for negative $d$s, as shown in Figure 2a. Thus, the favored weight distribution may be hard to obtain by tuning $d$.

To avoid such complications and alleviate the difficulties for manipulating the eigengraph weights, a transformation function $g(\cdot)$ operating on the graph Laplacian $\mathbf{L}_{sym}$ or $\tilde{\mathbf{L}}_{sym}$ is in need to shift $g(\lambda_i)$ or $g(\tilde{\lambda}_i)$ into a proper value range so that its power of a real-valued $d$ is easy to obtain and well-behaved versus $\lambda_i$ or $\tilde{\lambda}_i$. Without the loss of generality, our following analysis focuses on $\mathbf{L}_{sym}$ and $\lambda_i$. There may exist multiple choices for $g(\cdot)$ satisfying the requirements. In this work, we focus on the following transformation function:

$$\hat{\mathbf{S}} = g(\mathbf{L}_{sym}) = \frac{1}{2}(2\mathbf{I} - \mathbf{L}_{sym}). \quad (7)$$

This choice of $g(\cdot)$ holds three properties:
(1) *Positive eigenvalues.* Since we have $\mathbf{L}_{sym}$'s $i$-th eigenvalue $\lambda_i \in [0, 2)$, the corresponding eigenvalue of $\hat{\mathbf{S}}$ is $g(\lambda_i) = \frac{1}{2}(2 - \lambda_i) \in (0, 1]$. Thus, the $d$-th power of $g(\lambda_i)$ is computable for any $d \in \mathbb{R}$.
(2) *Monotonicity versus eigenvalues $\lambda$.* As shown in Figure 2b, $g(\lambda_i)^d = (1 - \frac{1}{2}\lambda)^d$ is monotonically increasing/decreasing when $\lambda$ varies between 0 and 2 under negative/positive depth. (3) *Geometric interpretability.* Filter $\hat{\mathbf{S}}$ can be expressed as:

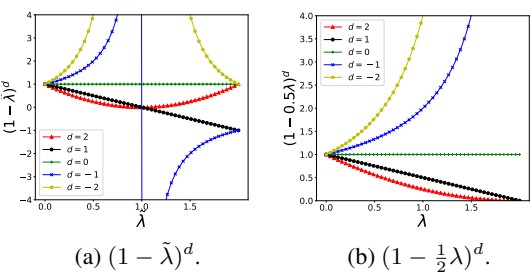

(a) $(1 - \tilde{\lambda})^d$.   (b) $(1 - \frac{1}{2}\lambda)^d$.

Figure 2: Eigengraph weight versus eigenvalue for (a) SGC and (b) TEDGCN-S under different depth $d$s.

$$\hat{\mathbf{S}} = \mathbf{U}\hat{\mathbf{\Lambda}}\mathbf{U}^\top = \mathbf{U}(\mathbf{I} - \frac{1}{2}\mathbf{\Lambda})\mathbf{U}^\top = \frac{1}{2}\mathbf{I} + \frac{1}{2}(\mathbf{I} - \mathbf{L}_{sym}) = \frac{1}{2}(\mathbf{I} + \mathbf{D}^{-\frac{1}{2}}\mathbf{A}\mathbf{D}^{-\frac{1}{2}}). \quad (8)$$

As shown in Figure 3, in the spatial domain, $\hat{\mathbf{S}}$ is obtained via 3 operations on adjacency matrix $\mathbf{A}$: normalization, adding self-loops, and scaling all edge weights by $\frac{1}{2}$ (a type of lazy random walk [39]), while $\tilde{\mathbf{S}}$ in vanilla GCN/SGC contains 2 operations: adding self-loops and normalization.

With the help of transformation $g$, the depth $d$ is redefined on real number domain, and the message propagation process of depth $d$ can be realized via the following steps: (1) Eigen-decompose $\mathbf{L}_{sym}$; (2) Calculate $\hat{\mathbf{S}}^d$ via weight $g(\lambda_i)^d$ and the weighted sum of all eigengraphs: $\hat{\mathbf{S}}^d = \mathbf{U}\hat{\mathbf{\Lambda}}^d\mathbf{U}^\top = \sum_{i=1}^n g(\lambda_i)^d \mathbf{u}_i\mathbf{u}_i^\top$ (3) Multiply $\hat{\mathbf{S}}^d$ with original node attributes $\mathbf{X}$.

**Negative depth explained.** First, an intuitive explanation of negative $d$ can be obtained from the perspective of matrix inverse and message diffusion process when $d$ takes integer values. Since $\hat{\mathbf{S}}^{-1}\hat{\mathbf{S}} = \mathbf{U}\hat{\mathbf{\Lambda}}^{-1}\mathbf{U}^\top\mathbf{U}\hat{\mathbf{\Lambda}}^1\mathbf{U}^\top = \mathbf{I}$, $\hat{\mathbf{S}}^{-1}$ is the inverse matrix of $\hat{\mathbf{S}}$. In diffusion dynamics, $\mathbf{X}$ can be viewed as an intermediate state generated in a series of message propagation steps. $\hat{\mathbf{S}}\mathbf{X}$ effectively propagates the message one-step forward, while $\hat{\mathbf{S}}^{-1}$ can cancel the effect of $\hat{\mathbf{S}}$ on $\mathbf{X}$ and recover the original message by moving backward: $\hat{\mathbf{S}}^{-1}\hat{\mathbf{S}}\mathbf{X} = \mathbf{X}$. Accordingly, $\hat{\mathbf{S}}^{-1}\mathbf{X}$ traces back to the message's previous state in

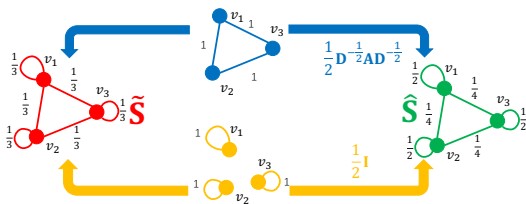

Figure 3: The difference between $\tilde{\mathbf{S}}$ (left) for GCN/SGC and $\hat{\mathbf{S}}$ (right) for TEDGCN.

the series. Non-integer $d$ indicates the back- or forward propagation can be a continuous process. Second, as the depth becomes negative, it primarily keeps the eigengraphs of high frequency (See Figure 2b), which have more negative edges. Therefore, it is equivalent to turning some positive edges in the original graph into negative edges in the augmented graph, which in turn makes node embeddings dissimilar in the message-passing process and thus naturally handles the heterophilic graphs (See Figure 1). More discussions on the impact of negative depth in spatial domain are presented in Section 4.4.

**TEDGCN-S algorithm.** By further making $d$ a trainable parameter, we present our model, *Trainable Depth-GCN-Single* (TEDGCN-S), whose final node embedding matrix is given by

$$\mathbf{H} = \sigma(\hat{\mathbf{S}}^d\mathbf{X}\mathbf{W}), \quad (9)$$

where $\sigma(\cdot)$ is the nonlinear activation function; $\mathbf{W}$ is a trainable parameter matrix; and $d$ is the trainable depth parameter. As observed from Figure 2b, weight distribution of different frequencies/eigengraphs is tuned via $d$: (1) for $d = 0$, the weight is uniformly distributed among all frequency components ($g(\lambda_i)^d = 1$), which implies that no particular frequency is preferred by the graph signal; (2) for $d > 0$, weight $g(\lambda_i)^d$ decreases with the corresponding frequency, which indicates the low frequency components are favored so that TEDGCN-S effectively functions as a low-pass filter and therefore captures graph homophily; (3) for $d < 0$, high frequency components gains amplified weights so that TEDGCN-S serves as a high-pass filter capturing graph heterophily. During training, TEDGCN-S tunes its frequency filtering functionality to suit the underlying graph signal by automatically finding the optimal $d$.

**TEDGCN-D algorithm.** During optimization, TEDGCN-S embraces a single depth $d$ unified for all eigengraphs and selects its preferences for either homophily or heterophily. However, TEDGCN-S requires a full eigen-decomposition of $\mathbf{L}_{sym}$, which can be expensive for large graphs. Additionally, the high and low frequency components in a graph signal may not be mutually exclusive, namely, there exists the possibility for a graph to simultaneously possess homophilic and heterophilic counterparts. Therefore, we propose the second variant of TEDGCN: TEDGCN-D (Dual), which introduces two separate trainable depths, $d_h$ and $d_l$, to gain more flexible weighting of the high and low frequency related eigengraphs respectively. Arnoldi method [30] is adopted to conduct EVD on $\mathbf{L}_{sym}$ and obtain the top-$K$ largest and smallest eigen-pairs $(\lambda_i, \mathbf{u}_i)$s. By denoting $\mathbf{U}_l = \mathbf{U}[:, 0 : K]$ and $\mathbf{U}_h = \mathbf{U}[:, n - K : n]$ ($\mathbf{U}_l$ and $\mathbf{U}_h \in \mathbb{R}^{n \times K}$), we define a new diffusion matrix $\hat{\mathbf{S}}_{dual}(d_l, d_h, K)$ as

$$\hat{\mathbf{S}}_{dual}(d_l, d_h, K) = \mathbf{U}_l \hat{\mathbf{\Lambda}}_l^{d_l} \mathbf{U}_l^\top + \mathbf{U}_h \hat{\mathbf{\Lambda}}_h^{d_h} \mathbf{U}_h^\top, \tag{10}$$

where $\hat{\mathbf{\Lambda}}_l \in \mathbb{R}^{K \times K}$ and $\hat{\mathbf{\Lambda}}_h \in \mathbb{R}^{K \times K}$ are diagonal matrices of the top-$K$ smallest and largest eigenvalues.[2] The final node embedding of TEDGCN-D is presented as

$$\mathbf{H} = \sigma(\hat{\mathbf{S}}_{dual}(d_l, d_h, K)\mathbf{X}\mathbf{W}), \tag{11}$$

where depths $d_l$ and $d_h$ are trainable; and $\mathbf{W}$ is a trainable parameter matrix. We make TEDGCN-D scalable on large graphs by choosing $K \ll n$, so that $\hat{\mathbf{S}}_{dual}(d_l, d_h, K)$ approximates the full diffusion matrix by covering only a small subset of all eigengraphs. For small graphs, we use $K = \lfloor \frac{n}{2} \rfloor$ to include all eigengraphs, and $\hat{\mathbf{S}}_{dual}(d_l, d_h, K)$ thus gains higher flexibility than $\hat{\mathbf{S}}$ with the help of the two separate depth parameters instead of a unified one.

**Complexity Analysis.** TEDGCN has two major steps: (1) conduct eigen-decomposition of $\hat{\mathbf{A}}$; (2) train a one-layer GCN on the augmented graph $\hat{\mathbf{S}}^d$. For Step (1), naive eigen-decomposition has a time complexity of $O(n^3)$. However, by adopting the Lanczos method [30] and only keeping the top-$K$ largest/smallest eigenvalues in TEDGCN-D, the time complexity can be reduced to $O(nK^2 + mK)$ [24], where $m$ is the number of edges and $K << n$. For Step (2), via dense matrix multiplications, computing the approximated $\hat{\mathbf{S}}^d$ takes $O(nK^2 + n^2K)$, and conducting message propagation takes $O(n^2q + nqc)$, where $q$ and $c$ denote the dimension of node feature vectors and the number of node classes. Potential improvements of efficiency: since the diffusion matrix in Subsection 4.4 of the augmented graph is sparse for known real-world graphs, the complexity of Step (2) can also be further simplified: the $O(n^2K)$ term can be reduced to $O(|E_{aug}|K)$, where $E_{aug}$ is the edge set of the augmented graph.

**Differences with ODE-based GNNs**. Previous attempts on GNNs with continuous diffusion are mostly inspired by graph diffusion equation, an Ordinary Difference Equation (ODE) characterizing the dynamical message propagation process versus time. In contrast, our framework starts from discrete graph convolution operations without explicitly involving ODE. CGNN [60] aims to build a deep GNN immune to over-smoothing by adopting the neural ODE framework [9]. But its time parameter $t$ is a non-trainable hyper-parameter predefined within the positive domain, which is the key difference with TEDGCN. A critical CGNN component for preventing over-smoothing, restart distribution (the skip connection from the first layer), is not needed in our framework. Moreover, CGNN applies the same depth to all frequency components, while TEDGCN-D has the flexibility to adopt two different depths respectively to be adaptive to high and low frequency components. GRAND [8] introduces non-Eular multi-step schemes with adaptive step size to obtain more precise solutions of the diffusion equation. Its depth (i.e., the total integration time) is continuous but still

---

[2]Case $\lambda_i = 2$, namely $g(\lambda_i) = 0$, is excluded since it corresponds to the existence of bipartite components.

predefined/non-trainable and only admits positive values. DGC [58] decouples the SGC depth into two predefined non-trainable hyper-parameters: a positive real-valued $T$ controlling the total time and a positive integer-valued $K_{dgc}$ corresponding to the number of diffusion steps. However, realizing negative depth in DGC is non-applicable since the implementation is through propagation by $K_{dgc}$ times, rather than through an arbitrary real-valued exponent $d$ on eigengraph weights in TEDGCN.

## 4 Experiment

In this section, we evaluate the proposed TEDGCN on the *semi-supervised* node classification task on both homophilic and heterophilic graphs.

### 4.1 Experiment Setup

**Datasets.** We use 11 datasets for evaluation, including 4 homophilic graphs: Cora [28], Citeseer [28], Pubmed [28] and DBLP [6], and 7 heterophilic graphs: Cornell [41], Texas [41], Wisconsin [41], Actor [41], Chameleon [44], Squirrel [44], and cornell5 [15]. We collect all datasets from the public GCN platform Pytorch-Geometric [15]. For Cora, Citeseer, Pubmed with data splits in Pytorch-Geometric, we keep the same training/validation/testing set split as in GCN [28]. For the remaining 8 datasets, we randomly split every dataset into 20/20/60% for training, validation, and testing.

**Baselines and metrics.** Here, we compare our model with 7 baseline methods, including 4 classic GNNs: GCN [28], SGC [59], APPNP [29] and ChebNet [12], and 3 GNNs tailored for heterophilic graphs: FAGCN [5], GPRGNN [10] and H2GCN [76]. Accuracy (ACC) is used as the evaluation metric. We report the average ACCs with the standard deviation (std) for all methods, each obtained by 5 runs with different initializations.

**Additional contents.** In Appendix, we provide more contents related to experiments including (1) the statistics of all datasets (Appendix A.1); (2) experimental results on synthetic graphs with *controllable edge homophily/heterophily levels* to demonstrate the effectiveness of negative depth (Appendix A.2); (3) comparison with additional methods including ACM-GCN [37, 38], LINKX [34], BernNet [22], GBK-GNN [14], HOG-GCN [57], Geom-GCN [41] and GloGNN [33] under the commonly used *fully-supervised* setting that training/validation/testing set split is 48/32/20% for training, validation, and testing based on the results reported in the original papers (Appendix A.3); (4) implementation details (Appendix A.4); and (5) visualization of node embeddings w.r.t. $d$ (Appendix A.5).

### 4.2 Node Classification

The semi-supervised node classification performances on homophilic graphs and heterophilic graphs are shown in Table 1 and Table 2 respectively.

**Homophilic graphs**. From Table 1, it is observed that different methods have similar performance on homophilic graphs. TEDGCN-S achieves the best accuracies on two datasets: Cora and DBLP. On the remaining two datasets, TEDGCN-S is only $1.1\%$ and $0.4\%$ below the best baselines (APPNP on Citeseer and SGC on Pubmed). For TEDGCN-D, it obtains similar performance as the other methods, even though it only uses the top-$K$ largest/smallest eigen-pairs.

Table 1: Performance comparison (mean±std accuracy (%)) on homophilic graphs.

| Datasets | Cora | Citeseer | Pubmed | DBLP |
|---|---|---|---|---|
| GCN | 80.8±0.8 | 70.5±0.6 | 78.8±0.6 | 84.1±0.2 |
| SGC | 80.9±0.4 | 70.8±0.8 | **79.6±0.4** | 84.1±0.2 |
| APPNP | 81.0±1.0 | **71.9±0.4** | 79.3±0.2 | 83.0±0.5 |
| GPRGNN | 82.0±0.7 | 69.3±0.9 | 78.6±0.7 | 84.5±0.3 |
| FAGCN | 80.3±0.4 | 71.7±0.8 | 78.5±0.9 | 82.4±0.7 |
| H2GCN | 78.8±1.0 | 70.5±1.0 | 77.9±0.3 | 82.4±0.3 |
| ChebNet | 78.8±0.5 | 71.1±0.4 | 78.1±0.8 | 83.1±0.1 |
| TEDGCN-S | **82.5±1.1** | 70.8±0.7 | 79.2±0.2 | **84.7±0.3** |
| TEDGCN-D | 82.4 ±0.7 | 70.6 ±0.6 | 77.9 ±0.3 | 84.2±0.2 |

**Heterophilic graphs**. Our TEDGCN-S or TEDGCN-D outperforms every baseline on all heterophilic graphs, as shown in Table 2. These results demonstrate that without manually setting the

Table 2: Performance comparison (mean±std accuracy (%)) on heterophilic graphs.

| Datasets | Texas | Cornell | Wisconsin | Actor | Squirrel | Chameleon | cornell5 |
|---|---|---|---|---|---|---|---|
| GCN | 55.9±3.4 | 44.3±4.4 | 51.4±2.2 | 27.5±0.5 | 35.8±1.3 | 55.2±1.8 | 67.9±0.2 |
| SGC | 58.7±3.1 | 43.8±4.4 | 47.3±2.1 | 28.0±0.8 | 37.2±1.8 | 55.3±1.0 | 67.4±0.5 |
| APPNP | 55.1±3.7 | 51.5±2.4 | 58.0±3.1 | 32.8±0.8 | 29.5±0.9 | 46.7±0.8 | 68.3±0.5 |
| GPRGNN | 61.3 ±5.8 | 53.3±4.6 | 71.0±4.8 | 33.6±0.4 | 34.1±1.0 | 55.0±3.9 | 67.3±0.3 |
| FAGCN | 60.2±7.8 | 54.8±7.4 | 60.1±5.2 | 32.3±0.5 | 31.2±1.6 | 50.4±1.9 | 68.3±0.7 |
| H2GCN | 68.8±6.5 | 61.4±4.4 | 69.9±5.3 | 33.9±0.3 | 30.4±0.9 | 48.8±1.9 | 68.4±0.2 |
| ChebNet | 76.2±2.9 | 66.7±3.9 | 75.4±3.5 | 34.3±0.5 | 31.8±0.5 | 49.6±1.8 | OOM |
| TEDGCN-S | **77.6±5.9** | 72.0±5.8 | **82.0±2.6** | **35.3±0.7** | 38.2±1.2 | 55.7±1.3 | 68.5±0.4 |
| TEDGCN-D | 77.1 ±2.5 | **72.0 ±2.8** | 81.5 ±2.4 | 27.6 ±0.8 | **44.2±0.9** | **56.9±0.9** | **70.0±0.2** |

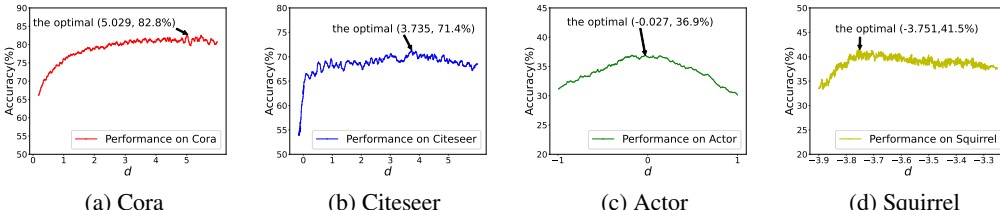

(a) Cora        (b) Citeseer        (c) Actor        (d) Squirrel

Figure 4: Node classification accuracy w.r.t. the trainable depth $d$ on four datasets: Cora, Citeseer, Actor and Squirrel. (the optimal $d$, accuracy) is annotated (e.g., (-0.027, 36.9%) for Actor).

model depth and without the prior knowledge of the input graph (i.e., whether the input graph is homophilic or heterophilic), TEDGCN has the capability of automatically detecting the underlying graph heterophily. We have an interesting observation: on 3 large datasets, Squirrel, Chameleon, and cornell5, even with only a small portion of the eigengraphs, TEDGCN-D is able to achieve better performance than TEDGCN-S with the complete set of eigengraphs. This suggests that the graph signal in some real-world graphs might be dominated by a few low and high frequency components, and allowing two independent depth parameters in TEDGCN-D brings the flexibility to capture the low and high frequencies at the same time.

## 4.3 Trainable Depth

A systematic study is conducted on the node classification performance w.r.t. the trainable depth $d$.
**Optimal depth**. In Figure 4, the optimal depths and their corresponding classification accuracies are annotated. For two homophilic graphs, Cora and Citeseer, the optimal depths are positive (5.029 and 3.735) in terms of the best ACCs, while for two heterophilic graphs, Actor and Squirrel, the optimal depths are negative ($-0.027$ and $-3.751$). These results demonstrate that our model can indeed automatically capture graph heterophily/homophily by finding the suitable depth to suppress or amplify the relative weights of the corresponding frequency components. Namely, high/low frequency components are suppressed for homophilic/heterophilic graphs respectively.

**Close to zero depth.** For the two homophilic graphs in Figures 4a and 4b, sharp performance drop is observed when depth $d$ approaches 0, since the eigengraphs gain close-to-uniform weights. For the heterophilic Actor dataset, its optimal depth $-0.027$ is close to 0, as shown in Figure 4c. In addition, the performance of TEDGCN-D (27.6%) is similar to that of GCN (27.5%), both of which are much worse than TEDGCN-S (35.3%). This result indicates that Actor is a special graph where all frequency compo-

Table 3: The performance of one-layer vanilla GCN over the augmented $\hat{\mathbf{S}}^d$.

| Datasets | Texas | Cornell | Wisconsin |
|---|---|---|---|
| GCN | 55.9 | 44.3 | 51.4 |
| TEDGCN-S | **77.6** | 72.0 | 82.0 |
| TEDGCN-D | 77.1 | 72.0 | 81.5 |
| GCN ($\hat{\mathbf{S}}^d$) | 75.9 | **72.7** | **83.4** |

nents have similar importance. Due to the absence of the intermediate frequency components between the high- and low-end ones, the performance of TEDGCN-D is severely impacted. For vanilla GCN, the suppressed weights of the high frequency components deviate from the near-uniform spectrum and thus lead to low ACC on this dataset.

## 4.4 Graph Augmentation and Geometric Insights

It is especially interesting to analyze what change a negative depth brings to the spatial domain and how such change impacts the subsequent model performance.

**Graph augmentation via negative depth.** By picking the optimal depth $d$ according to the best performance on the validation set, a new diffusion matrix $\hat{\mathbf{S}}^d$ is obtained. With the optimal $d$ fixed,

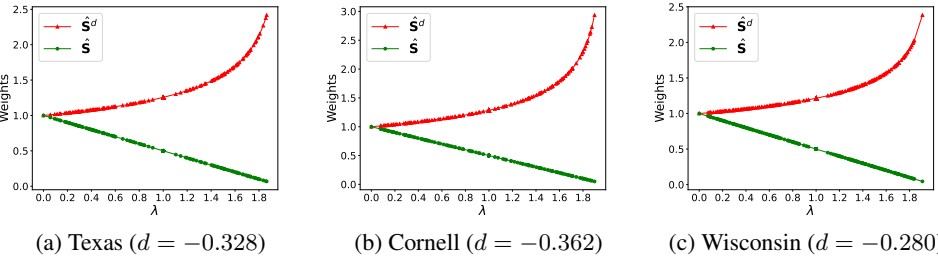

(a) Texas ($d = -0.328$)      (b) Cornell ($d = -0.362$)      (c) Wisconsin ($d = -0.280$)

Figure 5: The weights of eigengraphs w.r.t. eigenvalues on the augmented diffusion matrix $\hat{\mathbf{S}}^d$ and original $\hat{\mathbf{S}}$.

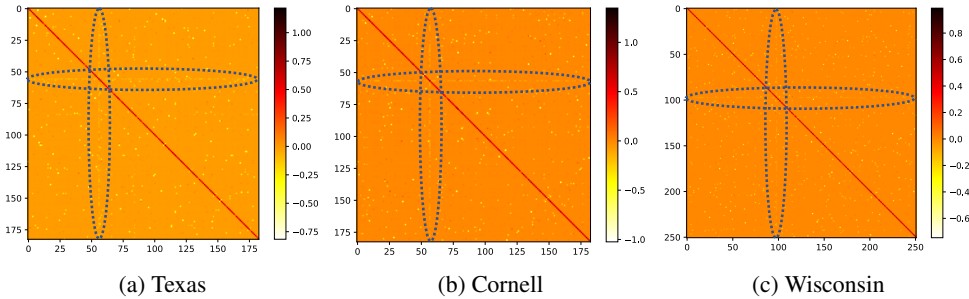

(a) Texas      (b) Cornell      (c) Wisconsin

Figure 6: The difference between the augmented diffusion matrix and the original one $\hat{\mathbf{S}}^d - \hat{\mathbf{S}}$.

substituting the normalized adjacency matrix $\tilde{\mathbf{D}}^{-\frac{1}{2}}\tilde{\mathbf{A}}\tilde{\mathbf{D}}^{-\frac{1}{2}}$ in Eq. (1) by $\hat{\mathbf{S}}^d$ is equivalent to applying the vanilla GCN to a new topology. This topology effectively plays the role of a structural augmentation for the original graph. The impact of such augmentation on performance is tested on 3 heterophilic graphs: Texas, Cornell and Wisconsin, as shown in Table 3. Apparently, for the vanilla GCN, the performance obtained with this new topology is superior over that with the raw input graph: it dramatically brings 20%-30% lifts in ACC. Moreover, the augmented topologies also make vanilla GCN outperform TEDGCN-S and TEDGCN-D on 2 out of the 3 datasets. By nature, the augmented graph is a re-weighted linear combination of the eigengraphs, and its topological structure intrinsically assigns higher weights to eigengraphs corresponding to higher frequencies, as shown in Figure 5.

**Geometric properties.** To further understand how the topology of $\hat{\mathbf{S}}^d$ with a negative optimal $d$ differs from that of $\hat{\mathbf{S}}$ and why the performance is significantly boosted, heat maps of $(\hat{\mathbf{S}}^d - \hat{\mathbf{S}})$s are presented in Figure 6. First, the dark red diagonal line in the heat map indicates the weights of self-loops are significantly strengthened in the augmented graph, and as a result, in consistency with the previous findings [71], the raw node attributes make more contributions in predicting their labels. These strengthened self-weights also play the similar role as restart distribution or skip connections [60] preventing the node embeddings becoming over-smoothed. In addition, there is a horizontal line and a vertical line (light yellow line marked by dashed ovals) in each heat map in Figure 6, correspond to the hub node in the graph, namely the node with the largest degree. Interestingly, the connections between this node and most other nodes in the graph experience a negative weight change. Therefore, the influence of the hub node on most other nodes are systematically reduced. Consequently, the augmentation amplifies the deviations between node embeddings and facilitates the characterization of graph heterophily.

## 5    Related Works

**Graph convolutional network (GCN).** GCN models can be mainly divided into two categories: (1) spectral graph convolutional networks and (2) spatial convolutional networks. In (1), Spectral CNN [7] borrows the idea from convolutional neural network [18] to construct a diagonal matrix as the convolution kernel. ChebNet [12] adopts a polynomial approximation of the convolution kernel. GCN [28] further simplifies the ChebNet via the first order approximation. Recently, [22, 4, 57] propose more advanced filters as the convolution kernel. Most works in (2) follow the *message-passing* mechanism. GraphSAGE [19] iteratively aggregates features from local neighborhood. GAT [50] applies self-attention to the neighbors. APPNP [29] deploys personalized pagerank [49] to sample nodes for aggregration. MoNet [40] unifies GCNs in the spatial domain.

**The depth of GCN and over-smoothing.** A large amount of works focus on the over-smoothing issue. Its intrinsic cause is demystified: a linear GCN layer is a Laplacian smoothing operator [32, 59]. PairNorm [69] forces distant nodes to be distinctive by adding an intermediate normalization layer. Dropedge [43], DeepGCN [31], AS-GCN [23], and JK-net [61] borrow the idea of ResNet [21] to dis-intensify smoothing. DeeperGXX [70] adopts a topology-guided graph contrastive loss for connected node pairs to obtain discriminative representations. Most works aim to build deep GCNs (i.e., $d$ is a large positive integer) by reducing over-smoothing, while TEDGCN extends the depth from $\mathbb{N}+$ to $\mathbb{R}$ and explores the negative depth.

**Node classification on homophilic and heterophilic graphs.** GCN/GNN models mostly follow the homophily assumption that connected nodes tend to share similar labels [28, 50, 19]. Recently, heterophilic graphs, in which neighbors often have disparate labels, attract lots of attention. Geom-GCN [41] and H2GCN [76] extend the neighborhood for aggregation. FAGCN [5] and GPRGNN [10] adaptively integrate the high/low frequency signals with trainable parameters. Alternative message-passing mechanisms have been proposed in HOG-GCN [57] and CPGNN [75]. The latest related works include ACM-GCN [37, 38], LINKX [34], BernNet [22], GloGNN [33] and GBK-GNN [14]. Other works can be found in a recent survey [72].

## 6    Conclusion and Future Work

To our best knowledge, this work presents the first effort to make GCN's depth become negative and trainable by redefining it on the real number domain. We unveil the intrinsic connection between negative GCN depth and graph heterophily. A novel problem of automatic GCN depth tuning for graph homophily/heterophily detection is formulated, and we propose a simple and powerful solution named TEDGCN with two variants (TEDGCN-S and TEDGCN-D). An effective graph augmentation method is also discussed via the new understanding on the message propagation mechanism generated by the negative depth. Superior performance of our method is demonstrated via extensive experiments with semi-supervised node classification on 11 graph datasets. The new insights on GCN's depth obtained by our work may open a new direction for future research on spectral and spatial GNNs. Our paper studies the depth of graph convolutional network and graph spectrum, which has no negative ethical impacts on society. Since TEDGCN requires to conduct eigen-decomposition of the graph Laplacian, it is not directly applicable to inductive and dynamic graph learning problems. In addition, there may exist other options for the transformation functions such as the pseudoinverse of the Laplacian. We leave these for future exploration.

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

# A Appendix

The appendix is structured as follows:

- Subsection A.1 presents the statistics of datasets used in the **Experiment** Section (Section 4) in the main contents.
- Subsection A.2 presents the experimental results on synthetic datasets with controllable homophily/heterophily level.
- Subsection A.3 compares TEDGCN with some latest methods on a different split (48%, 32%, 20%) for training, validation and testing of benchmark datasets. In this subsection, we also compare all methods (15 baselines and TEDGCN) under a unified setting on Texas, Cornell, Wisconsin and Actor datasets.
- Subsection A.4 introduces the details of the implementation of TEDGCN.
- Subsection A.5 is a visualization of node embeddings w.r.t. different depths ($d$ values) on both homophilic (Cora) and heterophilic (Texas) graphs.

## A.1 The Statistics of Datasets

Table 4: The statistics of datasets.

| Datasets | Cora | Citeseer | Pubmed | DBLP | Texas | Wisconsin | Cornell | Actor | Chameleon | Squirrel | cornell5 |
|---|---|---|---|---|---|---|---|---|---|---|---|
| #Nodes | 2,708 | 3,327 | 19,717 | 17,716 | 183 | 251 | 183 | 7,600 | 2,277 | 5,201 | 18,660 |
| #Edges | 10,556 | 9,104 | 88,648 | 105,734 | 325 | 515 | 298 | 30,019 | 62,792 | 396,846 | 158,1554 |
| #Features | 1,433 | 3,703 | 500 | 1,639 | 1,703 | 1,703 | 1,703 | 932 | 2,325 | 2,089 | 4,735 |
| #Classes | 7 | 6 | 3 | 4 | 5 | 5 | 5 | 5 | 5 | 5 | 5 |

## A.2 Experiments on Synthetic Datasets

Table 5: Performance comparison of TEDGCN and a 2-layer SGC on synthetic datasets. (%)

| Homophily Level $h(\mathcal{G})$ | 0.1 | 0.2 | 0.3 | 0.4 | 0.5 | 0.6 | 0.7 | 0.8 | 0.9 |
|---|---|---|---|---|---|---|---|---|---|
| SGC | 44.8 | 53.2 | 58.3 | 67.9 | 73.0 | 77.6 | 81.6 | 83.5 | 90.7 |
| TEDGCN | 67.3 | 68.5 | 60.3 | 72.3 | 78.3 | 78.6 | 82.7 | 84.0 | 91.5 |
| Optimal $d$ | -3.24 | -0.75 | -0.60 | 0.74 | 0.94 | 1.01 | 2.69 | 2.98 | 3.58 |

To better demonstrate the relation between the negative depth $d$ and edge homophily/heterphily, we construct synthetic datasets with different homophily levels. The synthetic graphs are generated via scheme adopted in ACM-GCN [38]. Specifically, there are 5 classes, each with 400 nodes. For a fixed homophily level $h(\mathcal{G})$ (e.g., $h(\mathcal{G}) = 0.1$), the node degree $\delta_v$ is sampled for each node via multinomial distribution. With the sampled $\delta_v$, $h(\mathcal{G})\delta_v$ intra-class edges and $(1 - h(\mathcal{G}))\delta_v$ inter-class edges are randomly generated. We compare TEDGCN with a two-layer SGC. The results on the synthetic datasets with different homophily levels are presented in Table 5. We can observe that (1) as the homophily level $h(\mathcal{G})$ decreases, the optimal $d$ turns from positive to negative; and (2) under low homophily levels, the performance of TEDGCN is significantly better than SGC.

## A.3 More Baselines and Unified Setting

Table 6: Performance comparison (mean±std accuracy)(%) on benchmark datasets.

| Datasets | Texas | Cornell | Wisconsin | Actor | Squirrel | Chameleon | Cora | Pubmed | Citeseer |
|---|---|---|---|---|---|---|---|---|---|
| BerNet | 82.70±2.70 | 81.35±4.05 | 87.05±2.54 | 34.46±0.77 | 34.02±1.14 | 47.01±1.60 | 87.51±0.63 | 85.35±0.57 | 76.82±0.94 |
| LINKX | 74.60±8.37 | 77.84±5.81 | 75.49±5.72 | 36.10 ±1.55 | 61.81±1.80 | 68.42±1.38 | 84.64±1.13 | 87.86±0.77 | 73.19±0.99 |
| GloGNN | 84.32±4.15 | 83.51±4.26 | 87.06±3.22 | 37.35±1.30 | 57.54±1.39 | **69.78±2.42** | 88.31±1.13 | 89.62±0.35 | 77.41 ± 1.65 |
| ACM-GCN | 87.84±4.40 | 85.14±6.07 | 88.43±3.22 | 36.63±0.84 | 55.19±1.49 | 69.14±1.91 | 87.91±0.95 | **90.00±0.52** | 77.32 ± 1.70 |
| TEDGCN-S | 90.32±4.71 | 85.81±4.21 | 88.63±4.51 | **37.40±1.11** | 48.43±1.42 | 67.02±1.91 | **88.41±1.73** | 85.34±2.11 | 77.42 ± 1.30 |
| TEDGCN-D | **91.41±3.62** | **86.53±3.80** | **91.42±4.22** | 29.03±1.03 | **62.02±3.03** | 67.33±1.71 | 87.90±1.31 | 84.84±1.21 | **77.81 ± 1.72** |

Under fully-supervised setting, we compare TEDGCN with ACM-GCN, LINKX, BernNet and GloGNN with the standard split of 48%/32%/20%. We directly use the performances of ACM-GCN, LINKX and GloGNN reported in the ACM-GCN paper. For BernNet, we obtain its performance by running its open-sourced code. The results are shown in Table 6. It is observed that the latest

methods, such as ACM-GCN and GloGNN do perform strongly. However, our model achieves the best classification accuracy on 7 out of the 9 datasets, which demonstrates the effectiveness of TEDGCN.

We also compare all methods (15 baselines and TeDGCN) under a unified setting. Since some methods do not report their performances on some datasets in their original paper and have not made their code publicly available, we compare all methods' performances on 4 commonly used heterophilic graph datasets: Texas, Cornell, Wisconsin and Actor, under the same public 48%/32%/20% split for training, validation and testing.

Table 7: Performance comparison (mean) on benchmark datasets.

| Datasets | Texas | Cornell | Wisconsin | Actor |
|---|---|---|---|---|
| GCN | 54.05 | 53.78 | 50.39 | 28.78 |
| SGC | 45.80 | 43.72 | 45.13 | 27.52 |
| GAT | 57.30 | 54.59 | 54.31 | 28.99 |
| APPNP | 58.92 | 51.12 | 58.84 | 31.83 |
| GPRGNN | 81.35 | 78.11 | 82.55 | 35.16 |
| FAGCN | 76.49 | 76.76 | 79.61 | 34.82 |
| H2GCN | 79.73 | 78.38 | 82.55 | 36.71 |
| ChebNet | 82.14 | 72.16 | 78.82 | 36.04 |
| BerNet | 82.70 | 81.35 | 87.05 | 34.46 |
| LINKX | 74.60 | 77.84 | 75.49 | 36.10 |
| GloGNN | 84.32 | 83.51 | 87.06 | 37.35 |
| ACM-GCN | 87.84 | 85.14 | 88.43 | 36.63 |
| GeomGCN | 66.76 | 60.54 | 64.51 | 31.59 |
| GBK-GNN | 70.27 | 75.68 | 68.63 | 36.12 |
| HOG-GCN | 85.17 | 84.32 | 86.67 | 36.82 |
| TeDGCN-S | 90.32 | 85.81 | 88.63 | **37.40** |
| TeDGCN-D | **91.41** | **86.53** | **91.42** | 29.03 |

From Table 7, we can observe that TeDGCN-S/TeDGCN-D outperforms all the remaining baselines.

### A.4  Implementation Details

We set the learning rate to $0.005$ or $0.01$, the decaying weight for the learning rate to $5e^{-4}$, the number of training epochs to 500. The $K$ largest/smallest eigenvalues for TEDGCN-D is $\max\{1000, 0.1n\}$, where $n$ is the number of nodes in the input graph. For all baselines, the parameters are set according to their original papers. All experiments are run on a Tesla-V100 GPU. Since this work is conducted during internship in Visa Research and the code can not be made public according to the policy of Visa Research. If you have any questions about the implementation, please email **yucheny5@illinois.edu** to get some help.

### A.5  Visualization of Node Embeddings w.r.t. Negative Depth $d$

We conduct a visualization of node embedding generated by TEDGCN under different depths for homophilic (Cora) and heterophilic (Texas) graphs. As shown in Figure 7, for Cora, when $d = 5$, the nodes in different classes form clearly distinguishable clusters (ACC=82.4%). When the depth decreases from a positive value $d = 5$ (ACC=82.4%) to a negative value $d = -2$ (ACC=29.5%), the nodes belonging to different classes/colors mix with each other, and the clearly clustered structure does not exist anymore. However, for Texas shown in Figure 8, when decreasing the depth from $d = 4$ (ACC=50%) to $d = -0.35$ (ACC=80%), the clusters of different classes/colors become well-separated. Especially, the purple nodes spread randomly all over the embedding space when $d = 4$ but concentrate in one cluster when $d = -0.35$. The spatial distribution of the node embeddings reveals that TEDGCN with positive and negative depths respectively models graph homophily and heterophily in a proper way.

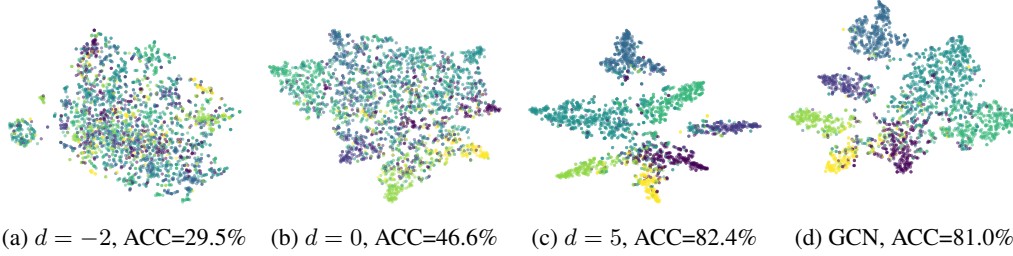

(a) $d = -2$, ACC=29.5%    (b) $d = 0$, ACC=46.6%    (c) $d = 5$, ACC=82.4%    (d) GCN, ACC=81.0%

Figure 7: Visualization of node embeddings by t-SNE on Cora (homophilic).

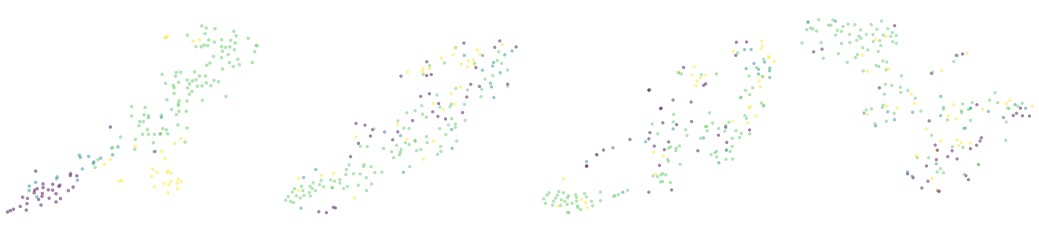

(a) $d = -0.35$, ACC=80.0%    (b) $d = 2$, ACC=59.0%    (c) $d = 4$, ACC=50.0%    (d) GCN, ACC=55.0%

Figure 8: Visualization of node embeddings by t-SNE on Texas (heterophilic).

