# OpenReview forum: "From Trainable Negative Depth to Edge Heterophily in Graphs"
_NeurIPS.cc/2023/Conference — NeurIPS 2023 poster_

### Official Review · Reviewer_jrHF · 2023-06-20

**Soundness:** 3 good
**Presentation:** 4 excellent
**Contribution:** 3 good
**Rating:** 7
**Confidence:** 4

**Summary:**

This paper tackles the problem of graph heterophily in training Graph Neural Networks. Through graph spectral analysis, the authors decompose the graph Laplacian matrix into eigengraphs. Dwelling on the observation that low/high-frequency eigengraphs correspond to homophily/heterophily, the authors propose to use a continuous and trainable depth parameter to model the weight of the eigengraphs so the trained GNN can be adaptive to the level of heterophily in the target graph. The authors also realize the efficiency issue of eigendecomposition in large graphs and hence propose an approximation method that only considers the top-k largest/smallest eigenvectors and two trainable depth parameters to improve the efficiency. Experimental results show promising improvement over baseline methods on heterophily datasets.

**Strengths:**

I enjoy reading the paper as the authors explain the details and intuition behind the equations very well. The experiments are also solid verification of the proposed methods. The functional calculus trick to transfer the depth from the integer domain to the real domain is interesting and novel. The approximation method TEDGCN-D seems like a good alternative to using the full eigendecomposition and is organically developed from the original version.

**Weaknesses:**

My primary concern lies in its comparison to existing work. The related work section only mentioned some work in the related field. However, a large body of work also realizes the importance of incorporating high-frequency passes in heterophilic graphs. How is the proposed method different from the existing ones? More specifically, note that in the formulation of the paper, the eigenvalues are shifted to (0,1] and d is uniform to both low/high frequency eigengraphs, meaning that d only shifts the focus on high/low frequency (amplifying the gap between eigenvalues too), what makes this different from other adaptive methods as the authors themselves point out in the paper?

**Questions:**

- line 187. I do not fully understand the negative depth explained for integer values. Since the proposed method only use either positive or negative depth, when does the model perform both positive message passing and negative message passing?
- The value of K is max(1000, 0.1n), I think 0.1n is still a very large value. (1) what is the processing time for TEDGCN-D on larger dataset, e.g. ogbn-arxiv. (2) What happens when a smaller K is used for a larger graph? Is the top few components sufficient?
- line 305. How is the optimal depth calculated? If it is a trainable parameter, how do you plot the full range of all depth?
- Is using a single depth parameter d necessary? For example, you can use trainable d for all different eigengraphs. Note that this should bring minimal extra computation cost as the complexity is still the same. This potentially provide more flexible adaptation to different graphs.

**Limitations:**

The proposed method uses spectrum and hence require additional computation for new and changed graphs. The cost of computing eigendecomposition for large graphs, even using the approximation method, is still unclear.

---

> ### Author Rebuttal · Authors · 2023-08-08
>
> **Q1: line 187. I do not fully understand the negative depth explained for integer values. Since the proposed method only use either positive or negative depth, when does the model perform both positive message passing and negative message passing?**
>
> **Response**: When the depth is negative, it does not mean **all** edges in the augmented graph are negatively weighted. As we point out in the analysis part, every graph can be regarded as a combination of many eigengraphs (Figure 1 and Eq. (3)). (1) Even the eigengraph corresponding to the highest frequency (e.g., the red eigengraph 3 in Figure 1) may still own some positively weighted edge (e.g. $\frac{1}{6}$ in eigengraph 3 in Figure 1). (2) After we obtain the negative depth, it is equivalent to merging all eigengraphs back into an augmented graph, where the eigengraph corresponding to the highest frequency may achieve large weight after merging. Due to reasons (1) and (2), these positively weighted edges in eigengraphs corresponding to high frequency may still be positively weighted, while some other edges may be changed from positively weighted to negatively weighted. In this case, there will exist both positive and negative edges in the augmented graph and the model performs both positive message passing and negative message passing simultaneously. This can also be verified in Figure 6: some red points represent that the original positive weights for some edges in the graph are still increased.
>
> **Q2 and Limitations: The value of K is max(1000, 0.1n), I think 0.1n is still a very large value. (1) what is the processing time for TEDGCN-D on larger dataset, e.g. ogbn-arxiv. (2) What happens when a smaller K is used for a larger graph? Is the top few components sufficient?**
>
> **Response**: It is true that 0.1n could be still large for a very large graph. (1) The processing time (most time is for the eigen-decomposition) for ogbn-arxiv is about 1 hour. (2) If the graph is only dominated by low/high frequency components, a smaller $K$ is sufficient for TEDGCN-D, which is validated by the results in Table 1 and Table 2. However, if the dataset contains non-trivial portion of intermediate frequency components, the top/bottom few components would not be sufficient, and TEDGCN-D may have a relatively poor performance, which can be observed in Table 2 on the Actor dataset: TEDGCN-S's accuracy (35.3\%) is much better than TEDGCN-D's (27.6\%). One possible solution for this problem (also a possible solution for the scalability issue) can be as follows: Since each node is closely related with the nodes within a multi-hop receptive field, we can divide the large graph into several connected components (i.e., subgraphs), set a maximum node number limit for each subgraph and run TEDGCN-S on each subgraph. For example, for a graph with 10000 nodes, we can divide it into 10 connected components/subgraphs and each subgraph has about 1000 nodes. In a vanilla implementation of eigen-decomposition with time complexity of $O(n^3)$, this solution can accelerate the algorithm on each subgraph by about 1000 times. In addition, this design will not sacrifice/discard low/high/intermediate frequency components. We leave the exploration to this tentative idea in future works.
>
> **Q3: line 305. How is the optimal depth calculated? If it is a trainable parameter, how do you plot the full range of all depth?**
>
> **Response**: Firstly, for each result in Tables 1 and 2, we set an initial depth value and then start training the model. During this process, the depth changes. When the best **validation accuracy** is reached, we compute the testing accuracy. Secondly, for Fig. 4, it is slightly different. Since it is only for illustration purpose, we skip the validation step and the initial unstable several training epochs. We directly plot the test accuracy that changes with the depth value on the fly during the training process, so that the plotted testing accuracy for the optimal depth in Fig. 4 is slightly higher than that in Tables 1 and 2. Different initial depths are tried, but the optimal values are very similar/close. Fig. 4 shows one realization in each case.
>
> **Q4: Is using a single depth parameter d necessary? For example, you can use trainable d for all different eigengraphs. Note that this should bring minimal extra computation cost as the complexity is still the same. This potentially provide more flexible adaptation to different graphs.**
>
> **Response**: This is a really good suggestion. It may open a door for broad future models. While single depth or dual depth are simple and effective, they are not the only strategies. For example, current TEDGCN-D can capture both low and high frequency components but may discard the intermediate frequency components. We may design models that can capture both low and intermediate frequency components or both high and intermediate frequency components in the future work. We may design models with three/four depths or even more complex/specific model with stronger expressive power in the spectrum domain.
>
> **Response to Weakness**: The graph spectrum analysis is indeed a commonly used tool in many works on handling heterophily. The differences between various models lies in the specific ways of manipulating the high/low-pass filters. As we describe in introduction, our **core theoretical contribution** can be summarized as below: To our best knowledge, we are the *first* to unveil the intrinsic relationship between the *negative GCN depth* and edge heterophily in graph. We also provide in-depth geometric and spectral explanations for negative depth.  We will include some discussions about this point in related works section in the revised version of our paper.

---

> > ### Comment · Reviewer_jrHF · 2023-08-12
> >
> > After reading the authors' response, I still believe this is a solid paper with a good contribution, and the answer cleared some of my concerns. I am happy to raise my score from 6 to 7.

---

> > > ### Author Response · Authors · 2023-08-12
> > > **Thanks**
> > >
> > > Thank you very much for reading our response and raising your score  !

---

### Official Review · Reviewer_iB1Z · 2023-07-07

**Soundness:** 3 good
**Presentation:** 3 good
**Contribution:** 2 fair
**Rating:** 6
**Confidence:** 4

**Summary:**

The paper proposes a new GCN that is capable of handling both homophilic and heterophilic networks. This method can be interpreted as generalizing the depth of the GCN, which is generally a natural number and a hyperparameter, to an arbitrary real number that is a trainable parameter. The first version of this method is essentially a variant of the well-known SGC method that makes the exponent on the graph matrix (corresponding the the depth) a trainable parameter. This can be computationally expensive, so the paper also proposes a second version which extracts the lowest frequency components and highest frequency components of the graph, and gives two tunable depth parameters to both sets of components. Experiments show that these versions generally achieve comparable or superior performance on the standard set of benchmark networks for homophilic/heterophilic node classification, relative to some prior methods. Some other experiments examine the depth parameters found by the method on various networks, as well as other insights from the method.

**Strengths:**

- The paper is organized well and easy to read.
- The proposed method is relatively simple as compared to similar recently proposed methods like GPR-GNN, but still demonstrates comparable or superior performance.
- The paper proposes and evaluates both a conceptually simple version of the method, as well as a slightly more complex but scalable version.

**Weaknesses:**

- As the authors note, this method is only directly applicable to the most basic setting of static undirected graphs in the transductive setting. Many of the competing methods to which this paper provides comparisons also apply to the inductive setting.

- While the simplicity of the method is appealing, since its dual variant (which is the only scalable variant) discards intermediate frequency components, it may struggle with graphs like Actor to which these components also seem crucial. The paper notes this.

- While the number of datasets is quite reasonable, and these have been the most popular ones in this field recently, I would have liked to see evaluations on different datasets. 3 of the datasets are very small, and two others (Squirrel/Chameleon) have recently been shown to have other issues - see the paper cited below. That paper also introduces other heterophilic datasets, which could yield a better evaluation.

- This is not really a weakness, but I do not see a significant conceptual contribution here prior to the introduction of the new method. There has been an abundance of recent papers on handling heterophily, and hence an abundance of discussion of various spectral filters and their interpretations.

### Typos/minor
- Line 101: "and the corresponding eigengraph $\mathbf{u}_1 \mathbf{u}_1^\top$ has an identical value $\frac{1}{n}$ for all entries" - isn't this only true for regular graphs?
- Line 125: "also corresponds a"
- Line 153-154: "applying an arbitrary function on a graph Laplacian is equivalent to applying the same function only on the eigenvalue matrix" - I think "arbitrary" has to be specified a bit here, e.g., this is true a matrix squaring function but not an entrywise squaring function
- Table 2 caption: "heterphilic"

**Questions:**

- The Squirrel and Chameleon datasets were recently found to have a lot of erroneous duplicates ("A critical look at the evaluation of GNNs under heterophily: are we really making progress?" by Platonov et al.). Is this paper using the revised datasets or the originals?

- The current method does not apply to the inductive setting, which is noted as a limitation of the work. Do the authors have any ideas on how the insights here could apply to that setting?

- The depth parameter found by the single variant often seems to be interpretable. Are the two depth parameters found by the dual variant also interpretable?

**Limitations:**

Limitations are very briefly discussed in the appendix and pertain to the limited setting to which the method is applicable (static/transductive). There could be more discussion of limitations beyond this, such as limitations about the evaluation.

---

> ### Author Rebuttal · Authors · 2023-08-08
>
> **Q1: The Squirrel and Chameleon datasets were recently found to have a lot of erroneous duplicates ("A critical look at the evaluation of GNNs under heterophily: are we really making progress?" by Platonov et al.). Is this paper using the revised datasets or the originals?**
>
> **Response**: We use the original datasets in our submitted paper. Thanks a lot for reminding us of the issues reported in this ICLR2023 paper. We will add this paper and some other works discussed in this paper into related works in the revised version of our paper. In addition, during the rebuttal period, we have evaluated TEDGCN-S on 5 new datasets provided in this ICLR 2023 paper by Platonov et al. (Revised Chameleon, Revised Squirrel, Tolokers, Minesweeper and Roman Empire) with same semi-supervised setting as other datasets in our paper. **The results are presented in Table 1 in the general response above** . We can observe that TEDGCN-S outperforms all baselines on Revised Chameleon, Revised Squirrel and Tolokers and achieves the second best performance on Minesweeper and Roman Empire, which demonstrates the effectiveness of TEDGCN-S on these 5 new datasets from Platonov et al.
>
> **Q2 and Weakness 1: The current method does not apply to the inductive setting, which is noted as a limitation of the work. Do the authors have any ideas on how the insights here could apply to that setting?**
>
> **Response**: For the inductive setting, one possible  solution is to borrow the idea from GraphSAGE about how to generalize the transductive GCN to the inductive setting. Following the similar idea as in GraphSAGE, since each node is closely related with the nodes within a multi-hop receptive field, we can sample a multi-hop subgraph (e.g.,3-5 hops) for the center node $v$ whose embedding needs to be optimized. Then, we can run TEDGCN-S on this small subgraph around the center $v$ to learn the optimal depth on this small subgraph. This design offers the flexibility for different subgraphs to learn different optimal depths. Furthermore, this can be applied to the inductive setting, when unseen/new node arrives, we can simply sample such a subgraph and run TEDGCN-S on this subgraph without re-training TEDGCN/conducting eigen-decomposition on the whole large graph, which can save a lot of time/space.
>
> **Q3: The depth parameter found by the single variant often seems to be interpretable. Are the two depth parameters found by the dual variant also interpretable?**
>
> **Response**: The two depths can be explained as follows: as we point out in the analysis part, every graph can be regarded as a combination of many eigengraphs (Figure 1 and Eq. (3)). TEDGCN-D will merge $K$ eigengraphs corresponding to the highest frequencies into a high-frequency augmented graph and merge $K$ eigengraphs corresponding to the lowest frequencies into another low-frequency augmented graph. Then, the high-frequency augmented graph and the low-frequency augmented graph will be considered together to learn the optimal combination of depths on both augmented graphs. The obtained depth on each augmented graph can be interpreted similarly as single depth in TEDGCN-S.
>
> **Response to Weakness 2**: It is true that TEDGCN-D may discard intermediate frequency components and this is reflected in the results of TEDGCN-D on Actor.  One possible solution for this problem (also a possible solution for the scalability issue) can be as follows: Since each node is closely related with the nodes within a multi-hop receptive field, we can divide the large graph into several connected components (i.e., subgraphs), set a maximum node number limit for each subgraph and run TEDGCN-S on each subgraph. For example, for a graph with 10000 nodes, we can divide it into 10 connected components/subgraphs and each subgraph has about 1000 nodes. In a vanilla implementation of eigen-decomposition with time complexity of $O(n^3)$, this solution can accelerate the algorithm on each subgraph by about 1000 times. In addition, this design will not sacrifice/discard low/high/intermediate frequency components. We leave the exploration to this tentative idea in future works.
>
> **Response to Weakness 3**: Thanks for the suggestion. Please refer to our response to **Q1** for this point.
>
> **Response to Weakness 4**: The graph spectrum analysis is indeed a commonly used tool in many works on handling heterophily. We will consider to move part of the background of spectrum graph theory to appendix in the revised version of our paper. Furthermore, as we describe in the introduction section, our core theoretical contribution can be summarized as below: To our best knowledge, we are the first to unveil the intrinsic relationship between the **negative GCN depth** and edge heterophily in graph. We also provide in-depth geometric and spectral explanations for negative depth.
>
> **Minors 1: Line 101: "and the corresponding eigengraph $\mathbf{u}\_{1}\mathbf{u}\_{1}^{\top}$
>  has an identical value $\frac{1}{n}$ for all entries" - isn't this only true for regular graphs?**
>
> **Response**: It is true for the symmetrically normalized graph Laplacian ($\mathbf{L}\_{sym}$) for **any** connected undirected graph. This is because any $\mathbf{L}\_{sym}=\mathbf{I} - \mathbf{D}^{-\frac{1}{2}}\mathbf{A}\mathbf{D}^{-\frac{1}{2}}$ for a connected undirected graph has a eigenvalue $0$, which has one corresponding eigengraph $\mathbf{u}\_{1}\mathbf{u}\_{1}^{\top}$.
>
> **Response to other minors/typos**: Thanks for the comments.  We will fix the typos and polish the text according to your suggestions in the revised version of our paper.

---

> > ### Comment · Reviewer_iB1Z · 2023-08-14
> >
> > Thank you for the response. I appreciate your running your method on the datasets from Platonov et al.
> >
> > Regarding Q3, apologies if my question was unclear. What I was looking for is not an explanation of the math of the two depth parameters, but rather something like Figure 4, where you show that for real-world network, with a single trainable depth, the depth that is found reflects whether the network is consider homophilous or heterophilous. Are two depth parameters similarly interpretable on real-world datasets?
> >
> > Regarding the eigenvector with zero eigenvalue:
> >
> > >Response: It is true for the symmetrically normalized graph Laplacian ($\mathbf{L}_\text{sym}$) for **any** connected undirected graph.
> >
> > According to [this](https://people.orie.cornell.edu/dpw/orie6334/Fall2016/lecture7.pdf), while the all-ones vector is a zero-eigenvalue eigenvector of the unnormalized Laplacian for any graph, for the normalized Laplacian, the eigenvector depends on the degree.

---

> > > ### Author Response · Authors · 2023-08-15
> > > **Further Discussion with Reviewer iB1Z**
> > >
> > > **Further P1: Regarding Q3, apologies if my question was unclear. What I was looking for is not an explanation of the math of the two depth parameters, but rather something like Figure 4, where you show that for real-world network, with a single trainable depth, the depth that is found reflects whether the network is consider homophilous or heterophilous. Are two depth parameters similarly interpretable on real-world datasets?**
> > >
> > > **Response**: Thanks for clarifying your question. For the real-world datasets: First, for homophilic datasets, similar results are obtained as TEDGCN-S, both $d\_{l}$ and $d\_{h}$ are positive, and $d\_{l}$ and $d\_{h}$ are close to the single $d$ in TEDGCN-S (e.g., $d\_{l}=5.65$ and $d\_{h}=5.98$ for Cora). Second, for heterophilic datasets, the results are a little bit different and interesting. For example, for Chameleon, $d\_{l}=1.56$ and $d\_{h}=-2.76$.
> > >
> > > This can be explained as the following: our adopted transformation function is $g(\lambda)=(1-0.5\lambda) \in (0, 1]$. When $d\ge 0$, $0<g(\lambda)^{d}\le 1$. When $d < 0$, $g(\lambda)^{d} > 1$, which is reflected in Figure 5. Assume that we have two eigenvalues $0<\lambda\_{l}<\lambda\_{h}<2$ and $d\_{h} < 0 < d\_{l}$. If we adopt one single negative depth $d\_h$, we have $g(\lambda\_{h})^{d\_h} > g(\lambda\_{l})^{d\_h} > 1 > g(\lambda\_{l})^{d\_l}$. Thus, to capture heterophily, a positive $d\_{l}$ and a negative $d\_{h}$ may further enlarge the relative weight strength between the high and low frequency eigengraphs (i.e., more flexibility), compared to a single negative depth. This is also validated in Table 2 that TEDGCN-D outperforms TEDGCN-S on Squirrel/Chameleon/cornell5. To conclude, because of the adopted transformation
> > > function ($g(\lambda)=(1-0.5\lambda) \in (0, 1]$), the weights of high frequency eigengraphs will be amplified to $>> 1$ with a negative $d\_{h}$ or reduced to $<< 1$ with a positive $d\_{h}$. So $d\_{h}$ plays an important role in adjusting the weights. According to the results of TEDGCN-D, positive $d\_{h}$ (with close positive $d\_{l}$) reflects homophily and negative $d\_{h}$ ($d\_{h} < 0 < d\_{l}$) reflects heterophily.
> > >
> > > **Further P2:Regarding the eigenvector with zero eigenvalue: Response: It is true for the symmetrically normalized graph Laplacian ($\mathbf{L}\_{sym}$) for any connected undirected graph. According to this, while the all-ones vector is a zero-eigenvalue eigenvector of the unnormalized Laplacian for any graph, for the normalized Laplacian, the eigenvector depends on the degree.**
> > >
> > >
> > > **Response**: Thanks for the reviewer pointing out that $\mathbf{u}\_{1}$ should be $\mathbf{D}^{0.5} \mathbf{e}$, where $\mathbf{e}$ is a all-ones vector. We will correct this point in the revised version of our paper. One thing we want to clarify is that when $\mathbf{u}\_{1} = \mathbf{D}^{0.5} \mathbf{e}$, the first eigengraph still corresponds to edge homophily in message-passing with all edges weighted positively, which will not affect the key idea and the model design of our work.

---

> > > > ### Comment · Reviewer_iB1Z · 2023-08-17
> > > >
> > > > Thank you for the response. I think it would be useful to add some discussion like the above of how the two depths reflect properties of each real-world network, perhaps with an accompanying plot or figure like Figure 4.
> > > >
> > > > Overall, while I still have some of the reservations from my original review, I appreciate the additions during the discussion period, in particular running experiments on the datasets from Platonov et al., so I have raised my score by a point.

---

> > > > > ### Author Response · Authors · 2023-08-17
> > > > > **Thanks**
> > > > >
> > > > > Thanks for the valuable suggestions, the detailed discussions and raising your score. We will include the above discussion about the interpretation of the two depths into the paper and polish the paper according to your suggestions in the revised version.

---

### Official Review · Reviewer_wZLL · 2023-07-07

**Soundness:** 4 excellent
**Presentation:** 4 excellent
**Contribution:** 4 excellent
**Rating:** 7
**Confidence:** 4

**Summary:**

This paper shows a connection between the depth of a GCN and it's suitability for homophilic / heterophilic graphs, by analyzing graph spectra. It proposes to address heterophily with negative depth and presents a GNN architecture called TeDGCN which allows for trainable and negative depth. TeDGCN outperforms / is competitive with other GCN variants on a number of graph datasets. It further leads to a graph augmentation approach when the optimal depth is known.

**Strengths:**

This paper tries to extend GCN with trainable real-valued depth. This has consequences for automatic tuning of the GCN depth, and for better addressing heterophilic datasets, both of which make it a significant novel contribution to the community. It establishes the connection between positive/negative depth with low/high frequency components of the graph signal and homophily/heterophily.

The analysis results in a new architecture TeDGCN. The paper further extends it to two variants, dealing with both low- and high-frequency components and addressing scalability challenges for large graphs. The method gives superior results, especially for heterophilic datasets. The trained depths are analyzed and also used to propose a graph augmentation mechanism which sometimes outperforms TeDGCN.

The paper is well-written and the ideas are presented clearly.

**Weaknesses:**

To enable trainable depth, the paper operates on a simplification of GCN which removes the intermediate non-linearities. Thus the resulting model is a linear model, which limits the expressive power of the approach. Also, as a GCN variant it is not applicable to the inductive setting.

It is not clear whether the lessons from this paper can be applied to more advanced GNNs or hold more broadly. Essentially trainable depth results in an augmented graph. Can this augmented graph be used with non-linear GNNs to still capture heterophily?

Some baselines with trainable depth (ODE-based GNNs) could be compared against. Is it possible to realize negative depth by inverting the graph Laplacian first?

**Questions:**

On line 262 the split should be 60/20/20 right?

Recent work [1] has shown pressing issues in old benchmarks of heterophilic datasets, which have been used here. Can you also include / replace with the results on the newer datasets proposed in [1]?

The same paper also shows that standard architectures like SAGE and GAT can outperform heterophily-specific architectures. Can you include these as baselines? Even if not adding datasets from [1], these baselines would be representative of models with higher expressive power than GCN or GCN-variants.

[1] Platonov et. al. "A critical look at the evaluation of GNNs under heterophily: Are we really making progress?" ICLR 2023.

**Limitations:**

The limitation to transductive problems is mentioned briefly in the appendix, but the limitation to linear models and expressive power is not discussed.

---

> ### Author Rebuttal · Authors · 2023-08-08
>
> **Q1: On line 262 the split should be 60/20/20 right?**
>
> **Response**: Thanks for your question. As we mentioned in line 254, we adopt the **semi-supervised** node classification task to evaluate TEDGCN and other baselines with the split 20/20/60\% for training/validation/testing set. The reason is that in real-world applications, it is very difficult to have the fully supervised setting with a large ratio of nodes' labels known and marked as training data points (e.g., 48\% or 60\%). Therefore, we believe the semi-supervised setting with much less than 50\% nodes functioning as training set is more reasonable to evaluate different methods, so without the loss of generality, we choose 20\%. For the fully supervised setting: (1) we have the results on benchmark datasets with the given 48/32/20\% split provided in [2] in Appendix A.5, which demonstrates the effectiveness of TEDGCN. Please refer to Appendix A.5 for more details. (2) We also evaluate TEDGCN on the given split of the revised chameleon dataset and the revised squirrel dataset provided by [1] during the rebuttal period. The accuracies for TEDGCN-S are $42.12 \pm 4.33$ (\%) on the revised chameleon and $41.70 \pm 1.97$ (\%) on the revised squirrel, both of which outperform all results from Table 2 in [1].
>
> [1] Platonov et. al. "A critical look at the evaluation of GNNs under heterophily: Are we really making progress?" ICLR 2023.
> [2] H. Pei, B. Wei, K. C.-C. Chang, Y. Lei, and B. Yang. Geom-gcn: Geometric graph convolutional networks. arXiv preprint arXiv:2002.05287, 2020.
>
> **Q2: Recent work [1] has shown pressing issues in old benchmarks of heterophilic datasets, which have been used here. Can you also include / replace with the results on the newer datasets proposed in [1]? The same paper also shows that standard architectures like SAGE and GAT can outperform heterophily-specific architectures. Can you include these as baselines? Even if not adding datasets from [1], these baselines would be representative of models with higher expressive power than GCN or GCN-variants.**
>
> **Response**: Thanks a lot for reminding us of this ICLR2023 paper, we will add this paper and some other works discussed in this paper into related works in the revised version of our paper. Following reviewer's suggestion, we have evaluated TEDGCN-S on 5 new datasets provided in [1] (Revised Chameleon, Revised Squirrel, Tolokers, Minesweeper and Roman Empire) with same semi-supervised setting as other datasets in our paper. We also include SAGE and GAT as baselines. **The results are as presented in Table 1 in the general response.** We can observe that (1) as mentioned in **Q2**, SAGE is a strong method and achieves the best accuracy on Minesweeper and Roman Empire; and (2) TEDGCN-S outperforms all baselines on Revised Chameleon, Revised Squirrel and Tolokers and achieves the second best performance on Minesweeper and Roman Empire, which demonstrates the effectiveness of TEDGCN-S on these 5 new datasets in [1].
>
> **Other questions in Weaknesses/Comments/Limitations**:
>
> **P1: The resulting model is a linear model, which limits the expressive power of the approach. Also, as a GCN variant it is not applicable to the inductive setting.**
>
> **Response**: (1) For the limited expressive power of linear model, first, our results show that with a proper depth, nonlinearity may be not be a critical factor to determine the performance, and the simplicity is a natural byproduct of our framework which requires no sacrifice of the performance. Second, one possible solution for this question is to leverage the augmented graph obtained by TEDGCN as the input for other non-linear GNNs, in which we can add non-linearity between each layer. (2) For the inductive setting, one possible solution is to borrow the idea from (Graph)SAGE about how to generalize the transductive GCN to the inductive setting. Following the same idea in (Graph)SAGE, since each node is closely related with the nodes within a multi-hop receptive field, we can sample a multi-hop subgraph (e.g.,3-5 hops) for the center node $v$ whose embedding needs to be optimized. Then, we can run TEDGCN-S on this small subgraph around the center $v$ to learn the optimal depth on this small subgraph. This design offers the flexibility for different subgraphs to learn different optimal depths. Furthermore, this can be applied to the inductive setting, when unseen/new node arrives, we can simply sample such a subgraph and run TEDGCN-S on this subgraph without re-training TEDGCN/conducting eigen-decomposition on the whole large graph, which can save a lot of time. We will leave the implementation of this idea to future works.
>
> **P2: Essentially trainable depth results in an augmented graph. Can this augmented graph be used with non-linear GNNs to still capture heterophily?**
>
> **Response**: Yes, the augmented graph can be used with non-linear GNNs, which only need to treat the augmented graph as input. One example is that in Subsection 4.4, we use the augmented graph as input for normal (non-linear) GCN, which achieves even better performance than TEDGCN-S/TEDGCN-D in capturing heterophily on the Cornell dataset and the Wisconsin dataset.
>
> **P3: Is it possible to realize negative depth by inverting the graph Laplacian first?**
>
> **Response**: We are afraid that this is not mathematically feasible. Actually, we do have thought about directly inverting the graph Laplacian first. However, the graph Laplacian for any connected graph has an eigenvalue 0, which makes the graph Laplacian non-invertible.

---

> > ### Comment · Reviewer_iB1Z · 2023-08-14
> >
> > Minor comment regarding the following:
> >
> > >We are afraid that this is not mathematically feasible. Actually, we do have thought about directly inverting the graph Laplacian first. However, the graph Laplacian for any connected graph has an eigenvalue 0, which makes the graph Laplacian non-invertible.
> >
> > It is possible to use a pseudoinverse of the Laplacian instead. This is a well-studied technique, see, e.g., [this link](https://arxiv.org/abs/2109.14587).

---

> > > ### Author Response · Authors · 2023-08-15
> > > **Response to the minor comment by Reviewer iB1Z**
> > >
> > > **Response**: Since the graph Laplacian for any connected graph is non-invertible, it is natural to come up with the idea of conducting the proposed transformation in the paper, which enjoys (1) positive eigenvalues; (2)monotonicity versus eigenvalues; and (3) good geometric interpretability.
> > >
> > > We thank the reviewer for pointing out the pseudoinverse technique.  We agree with the reviewer that by generalizing the **inverse requirement to pseudoinverse technique**, it is possible to realize negative depth by inverting the graph Laplacian first. To better answer reviewer's question, we have tried a simple implementation of direct pseudoinverse of the symmetrically normalized graph Laplacian $\mathbf{L}\_{sym}$ on the revised squirrel and the revised chameleon with same semi-supervised setting in our paper. The ACC for the revised chameleon is 37.07\% and for the revised squirrel is 34.71\%, both of which are worse than TEDGCN-S(40.03\% for the revised chameleon and 37.21\% for the revised squirrel). One possible reason is that the direct pseudo-inverse might own some different properties from the **real** inverse/the transformation function. We leave the detailed analysis and improvement of the pseudoinverse technique as the future work.

---

> > > > ### Comment · Reviewer_wZLL · 2023-08-19
> > > >
> > > > I thank the authors for their response. I have updated my rating. Please make sure to include the results / experiments discussed in the rebuttal (such as the pseudoinverse one).

---

### Official Review · Reviewer_yXi3 · 2023-07-09

**Soundness:** 3 good
**Presentation:** 4 excellent
**Contribution:** 2 fair
**Rating:** 6
**Confidence:** 3

**Summary:**

This paper proposes two algorithms to learn GNN with a trainable depth. At first, the authors exploit previous theoretical results about the correlation between frequency and zero crossings and the intuition that capturing heterophily needs more zero crossings to motivate adjusting the weights of eigen-graphs. Then it is natural to achieve this goal by tuning the depth of a GNN, where the authors propose to extend the possible space of depths to real numbers. To my knowledge, it is the first attempt to train a GNN with trainable and possibly negative depth, especially since the proposed algorithms are not built upon continuous-time diffusions. Extensive experiments are conducted, showing that the proposed algorithms are quite effective for heterophilic graphs, and they tend to learn the optimal depth.

**Strengths:**

1. This well-written paper enables me to pick up its core ideas effortlessly. Particularly, the provided examples are quite helpful for understanding the corresponding definitions and formulas.
2. Learning a negative depth to encourage larger weights for eigen-graphs that provide more zero crossing interests me. More importantly, it is well-motivated and demonstrates significant advantages in capturing heterophilic graph patterns.
3. The empirical studies seem to be comprehensive. At first, both homophilic and heterophilic graphs are included. I am glad to see that the authors honestly report the results on those four homophilic graphs, where the proposed algorithms have not achieved a consistent improvement. And it is great to see the dramatic improvements on heterophilic graphs, which strongly support the idea of this paper. Second, the detailed analysis confirms that the proposed algorithms can truly learn the optimal depth. This lets us attribute the successes in capturing heterophilic graph patterns to such a mechanism and its consequence of up-weighting zero crossings.
4. I am quite fond of the interpretation of augmenting original graph structure, which perfectly obeys our intuition.

**Weaknesses:**

1. How to apply the proposed algorithms to large graphs is a critical question that has not been fully resolved.
2. The motivation of proposed algorithms seems to be entirely built upon previous theoretical results. I am curious about what is exactly the contribution of this paper.
3. To achieve a trainable and negative depth, the designed GNN is in a similar form to SGC, which also sacrifices some non-linearity (or expressiveness) to gain simplicity.
4. The rescaling of the base into (0, 1) is reasonable and explainable, which is satisfactory to me. Yet, it would be better to justify its expressiveness and numerical properties as original bases (i.e., original frequencies).

**Questions:**

In line 130, the cardinalities of examples are 1 and 2. As the considered graph is undirected, why not 2 and 4?

**Limitations:**

There is a complexity analysis of the proposed algorithms, which can be regarded as a discussion of the limitations.

---

> ### Author Rebuttal · Authors · 2023-08-08
>
> **Q1: In line 130, the cardinalities of examples are 1 and 2. As the considered graph is undirected, why not 2 and 4?**
>
> **Response**: Thank you for raising this question. In undirected graph, we regard edge $(v\_{j}, v\_{k})$ and edge $(v\_{k}, v\_{j})$ as one edge. We will add a footnote here to clarify this point in the revised version of this paper.
>
>
> **Other points in Weaknesses/Comments/Limitations**:
>
> **P1:How to apply the proposed algorithms to large graphs is a critical question that has not been fully resolved.**
>
> **Response**: We think that the TEDGCN-D in the paper is proposed from the *spectral domain* to reduce the time complexity on large graphs. An alternative solution is from the *spatial domain*: Since each node is closely related with the nodes within a multi-hop receptive field, we can divide the large graph into several connected components (i.e., subgraphs), set a maximum node number limit for each subgraph and run TEDGCN-S on each subgraph. For example, for a graph with 10000 nodes, we can divide it into 10 connected components/subgraphs and each subgraph has about 1000 nodes. In a vanilla implementation of eigen-decomposition with time complexity of $O(n^3)$, this solution can accelerate the algorithm on each subgraph by about 1000 times. We leave the exploration to this tentative idea in future works.
>
> **P2: The motivation of proposed algorithms seems to be entirely built upon previous theoretical results. I am curious about what is exactly the contribution of this paper.**
>
> **Response**: The graph spectrum analysis is indeed a commonly used tool in many works (e.g.,GCN). We look at the the existing theory from a novel angle and our core contributions can be summarized as below: (1) From the theoretical aspect: to our best knowledge, we are the first to unveil the intrinsic relationship between the negative GCN depth and edge heterophily in graph. We also provide in-depth geometric and spectral explanations for negative depth. (2) From the model aspect, we propose a simple and powerful model TEDGCN with two variants (TEDGCN-S and TEDGCN-D) and discuss a novel graph augmentation method based on TEDGCN.
>
> **P3: To achieve a trainable and negative depth, the designed GNN is in a similar form to SGC, which also sacrifices some non-linearity (or expressiveness) to gain simplicity.**
>
> **Response**: First, our results show that with a proper depth, nonlinearity may be not be a critical factor to determine the performance, and the simplicity is a natural byproduct of our framework which requires no sacrifice of the performance. Second, as pointed by Reviewer `wZLL`, one possible solution for this question is to leverage the augmented graph obtained by TEDGCN as the input for other non-linear GNNs, in which we can add non-linearity between each layer.
>
> **P4: The rescaling of the base into (0, 1) is reasonable and explainable, which is satisfactory to me. Yet, it would be better to justify its expressiveness and numerical properties as original bases (i.e., original frequencies).**
>
> **Response**: Actually, we do have thought about using the original frequencies in TEDGCN. However, every connected graph has an eigenvalue 0, which will lead to problem when we have negative depth (e.g., -1) because $0^{-1} = \frac{1}{0}$ is meaningless, and this makes it hard to explain the numerical properties in the original spectrum. Thus, we propose a transformation function $g(\lambda)=1-\frac{1}{2}\lambda\in (0,1]$ to solve this problem.

---

> > ### Comment · Reviewer_yXi3 · 2023-08-13
> > **Discussions**
> >
> > Thanks for your response! Most of my concerns are resolved. I will keep my score to support this paper.

---

> > > ### Author Response · Authors · 2023-08-13
> > > **Thanks**
> > >
> > > Thanks for reading our response and supporting our paper!

---

### Author Rebuttal · Authors · 2023-08-08

**General Response for All Reviewers**:

We sincerely thank all reviewers for their valuable time and insightful feedbacks which are very helpful for further improving the quality of our paper. We are grateful that the reviewers appreciate the novelty of our work (`wZLL`:"a significant novel contribution to the community", `yXi3`:"to my knowledge, it is the first attempt", `jrHF`: "interesting and novel"). We are also encouraged that reviewers think that (1) our paper is well-written and the key idea is presented clearly and easy to follow (`yXi3`, `wZLL`, `iB1Z` and `jrHF`); and (2) the empirical experiments are comprehensive (`yXi3`) and solid (`jrHF`). We have provided our point-to-point response to the questions of each reviewer below and we sincerely invite the reviewers for further discussion.

In addition, during the rebuttal period, as suggested by Reviewer `wZLL` and Reviewer `iB1Z`,  we have evaluated TEDGCN-S on 5 new datasets provided in the ICLR 2023 paper by Platonov et al [1]. (Revised Chameleon, Revised Squirrel, Tolokers, Minesweeper and Roman Empire) with same semi-supervised setting as other datasets in our paper. The results are as presented in Table 1.

[1] Platonov et. al. "A critical look at the evaluation of GNNs under heterophily: Are we really making progress?" ICLR 2023.

Table 1: Performance comparison on 5 new datasets in [1] ($ACC \pm std$(\%))

| Model | Revised Chameleon | Revised Squirrel | Tolokers | Minesweeper | Roman Empire|
| ------------ | :-----------: | :-----------: | :-----------: | :-----------: | :-----------: |
|  SGC  | $35.92 \pm 2.06$ | $29.70 \pm 1.25$ | $78.31 \pm 0.64$ | $80.02 \pm 0.39$ | $39.01 \pm 0.78$ |
|  GCN  | $36.70 \pm 2.72$ | $30.75 \pm 2.09$ | $78.49 \pm 0.51$ | $79.92 \pm 0.41$ | $39.27 \pm 0.42$ |
|  SAGE  | $35.43 \pm 3.80$ | $31.75 \pm 1.67$ | $\textit{78.54} \pm \textit{0.65}$ | $\textbf{82.22} \pm \textbf{0.32}$ | $\textbf{65.57} \pm \textbf{0.92}$ |
|  GAT  | $36.59 \pm 1.18$ | $31.53 \pm 0.89$ | $78.01 \pm 0.51$ | $80.02 \pm 3.30$ | $38.71 \pm 0.98$ |
|  APPNP  | $37.00 \pm 2.18$ | $31.91 \pm 1.19$ | $78.05 \pm 0.49$ | $80.04 \pm 0.37$ | $62.03 \pm 0.41$ |
|  GPRGNN  | $\textit{37.08} \pm \textit{1.88}$ | $\textit{36.60} \pm \textit{0.73}$ | $78.09 \pm 0.51$ | $80.06 \pm 0.78$ | $57.28 \pm 0.49$ |
|  FAGCN  | $35.96 \pm 2.61$ | $32.50 \pm 4.58$ | $78.37 \pm 0.54$ | $80.11 \pm 0.40$ | $62.81 \pm 0.89$ |
|  TEDGCN-S  | $\textbf{40.03} \pm \textbf{1.60}$ | $\textbf{37.21} \pm \textbf{1.70}$ | $\textbf{78.58} \pm \textbf{0.78}$ | $\textit{80.41} \pm \textit{0.70}$ | $\textit{63.05} \pm \textit{0.96}$ |

We can observe that TEDGCN-S outperforms all baselines on Revised Chameleon, Revised Squirrel and Tolokers and achieves the second best performance on Minesweeper and Roman Empire, which demonstrates the effectiveness of TEDGCN-S on these 5 new datasets from Platonov et al.

---

### Decision · Program_Chairs · 2023-09-21

**Decision:**

Accept (poster)

**Comment:**

The paper introduces a novel approach to adaptively train Graph Neural Networks (GNNs) by proposing trainable depth parameters to handle graph heterophily. The theory is based on graph spectral analysis and supported by exhaustive experiments. Reviewers value the presentation's clarity, innovative contribution, and empirical validity.

While reviewers acknowledge the strengths, they do raise some considerations. These include the applicability of the approach to advanced GNNs, potential limited expressiveness attributed to linearity, and the concentration on transductive settings. However, the substantive value of contributions and the comprehensive evaluation outweigh these concerns. Therefore, the paper is recommended for acceptance. The authors are suggested to incorporate the reviewers' feedback during the preparation of the final version of the paper.